METHODS AND RESOURCES

# Single-cell transcriptomic analysis reveals rich pituitary–Immune interactions under systemic inflammation

Ting Yan[1,2,3☉], Ruiyu Wang[2,3,4☉], Jingfei Yao[3], Minmin Luo🔵[2,3,5,6,7]*

1 School of Life Sciences, Tsinghua-Peking Center for Life Sciences, Tsinghua University, Beijing, China, 2 Chinese Institute for Brain Research, Beijing, China, 3 National Institute of Biological Sciences (NIBS), Beijing, China, 4 PTN Graduate Program, School of Life Sciences, Peking University, Beijing, China, 5 Tsinghua Institute of Multidisciplinary Biomedical Research (TIMBR), Beijing, China, 6 New Cornerstone Science Laboratory, Shenzhen, China, 7 Research Unit of Medical Neurobiology, Chinese Academy of Medical Sciences, Beijing, China

☉ These authors contributed equally to this work.
* luominmin@cibr.ac.cn

**Data Availability Statement:** All data generated or analyzed during this study are included in this published article (and its supporting information files). All fastq files, expression matrix and metadata table are available from the National

## Abstract

The pituitary represents an essential hub in the hypothalamus–pituitary–adrenal (HPA) axis. Pituitary hormone-producing cells (HPCs) release several hormones to regulate fundamental bodily functions under normal and stressful conditions. It is well established that the pituitary endocrine gland modulates the immune system by releasing adrenocorticotropic hormone (ACTH) in response to neuronal activation in the hypothalamus. However, it remains unclear how systemic inflammation regulates the transcriptomic profiles of pituitary HPCs. Here, we performed single-cell RNA-sequencing (scRNA-seq) of the mouse pituitary and revealed that upon inflammation, all major pituitary HPCs respond robustly in a cell type-specific manner, with corticotropes displaying the strongest reaction. Systemic inflammation also led to the production and release of noncanonical bioactive molecules, including Nptx2 by corticotropes, to modulate immune homeostasis. Meanwhile, HPCs up-regulated the gene expression of chemokines that facilitated the communication between the HPCs and immune cells. Together, our study reveals extensive interactions between the pituitary and immune system, suggesting multifaceted roles of the pituitary in mediating the effects of inflammation on many aspects of body physiology.

## Introduction

As the master gland of the endocrine system and a key component in the hypothalamus–pituitary–adrenal (HPA) axis, the pituitary releases several hormones to regulate numerous physiological functions, such as development, sexual maturation, reproduction, gestation, metabolism, lactation, and stress handling [1]. It also participates in controlling immune responses to inflammatory stimuli [2]. During systemic inflammation that is caused by bacterial or viral infections such as Severe Acute Respiratory Syndrome Coronavirus 2 (SARS-CoV-2) [3–5], the HPA axis orchestrates the release of glucocorticoids to restrain inflammation [6].

Genomics Data Center (NGDC) under the accession number PRJCA015861. The pipeline for processing the STRT-seq2 data was written in Snakemake and deposited at https://github.com/RuiyuRayWang/ScRNAseq_smkpipe_at_Luolab. Additional codes for analyzing the data were deposited at https://github.com/RuiyuRayWang/pituitary_inflammation.

**Funding:** This work was supported by Ministry of Science and Technology STI2030-Major Projects (2021ZD0202803), the Research Unit of Medical Neurobiology at Chinese Academy of Medical Sciences (2019RU003), Beijing Municipal Government, Tsinghua University and New Cornerstone Science Foundation to M.L. The funder had no role in study design, data collection and analysis, the decision to publish, or the preparation of the manuscript.

**Competing interests:** The authors have declared that no competing interests exist.

**Abbreviations:** AAV, adeno-associated virus; BMDM, bone marrow-derived macrophage; CLP, cecal ligation and puncture; CRH, corticotropin-releasing hormone; DEG, differentially expressed gene; DPBS, Dulbecco's phosphate-buffered saline; FA, formic acid; FACS, fluorescence-activated cell sorting; FSH, follicle-stimulating hormone; GH, growth hormone; GO, Gene Ontology; GSEA, Gene Set Enrichment Analysis; HPA, hypothalamus–pituitary–adrenal; HPC, hormone-producing cell; HS, horse serum; HVG, highly variable gene; ISH, in situ hybridization; LH, luteinizing hormone; LPS, lipopolysaccharide; MAD, median absolute deviation; M-CSF, macrophage colony-stimulating factor; MSH, melanocyte-stimulating hormone; MWCO, molecular weight cut-off; NBR, neutrophil-to-B cell ratio; NLR, neutrophil-to-lymphocyte ratio; PCA, principal component analysis; PBMC, peripheral blood mononuclear cell; PRL, prolactin; PVN, paraventricular nucleus; RBC, red blood cell; SARS-CoV-2, Severe Acute Respiratory Syndrome Coronavirus 2; SCENIC, single-cell regulatory network inference and clustering; scRNA-seq, single-cell RNA-sequencing; STRT-seq, single-cell tagged reverse transcription sequencing; TNF-α, tumor necrosis factor α; TSH, thyroid-stimulating hormone; UMAP, uniform manifold approximation and projection; UMI, unique molecular identifier; WBC, white blood cell.

The current model suggests that, upon peripheral immune stimulation, activated immune cells produce proinflammatory cytokines, which stimulate vagus nerve activity and in turn activate the paraventricular nucleus (PVN) in the hypothalamus [7,8]. Activation of PVN neurons leads to the release of corticotropin-releasing hormone (CRH) into the anterior pituitary, where corticotropes are triggered to release ACTH [9]. Through general circulation, ACTH travels to the adrenal glands to initiate steroidogenesis and the release of glucocorticoids, which finally prevent the immune system from overreaction [10,11].

Several unresolved questions pertain to interactions between the pituitary and the immune system. The pituitary gland comprises 6 major types of hormone-producing cells (HPCs): somatotropes (producing the growth hormones-GH), corticotropes (producing ACTH), lactotropes (producing prolactin-PRL), gonadotropes (producing luteinizing hormone-LH and follicle-stimulating hormone-FSH), thyrotropes (producing thyroid-stimulating hormone-TSH), and melanotropes (producing melanocyte-stimulating hormone-MSH) [12]. The specific responses of diverse pituitary HPCs to inflammatory stimuli remain uncertain. Clarifying this issue could shed light on how immune stress influences the hormonal regulation of various physiological functions. Additionally, aside from conventional hormones, pituitary HPCs might secrete other factors. Identifying these factors could expand our understanding of HPC functions. Lastly, it remains unclear how direct pituitary–immune communication impacts pituitary physiology.

The emergence of the single-cell RNA sequencing (scRNA-seq) technique has enabled comprehensive characterization of pituitary transcriptional responses in a cell type-specific manner. While prior studies have illuminated the pituitary's transcriptional landscape in physiological conditions [13–16], systematic investigation into pituitary HPCs' transcriptomics under systemic inflammatory challenges remains absent. In this study, we performed scRNA-seq of the pituitary glands of mice subjected to systemic immune stimuli. Our findings unveil robust transcriptional responses across all pituitary HPCs in the face of inflammatory challenges, with corticotropes exhibiting the most pronounced reaction. Systemic inflammation potently induces the pituitary expression of many genes encoding secreted factors like cytokines. This approach also led to the identification of Nptx2, a secreted protein from corticotropes, serving as a pituitary inflammation biomarker. Combining cell–cell communication analysis of the transcriptomics with experimental validations revealed extensive crosstalk between pituitary cells and immune cells during inflammation. Together, these results reveal previously underappreciated interactions between the pituitary gland and immune system under inflammatory conditions, highlighting potent effects of immune challenges on a range of HPA-related, fundamental physiological responses.

## Results

### Transcriptomic characterization of the pituitary under systemic inflammation

To explore the response of the pituitary cells to immune challenges, we first performed bulk RNA sequencing on pituitary tissues extracted from mice that were intraperitoneally (i.p.) injected with a medium-dose lipopolysaccharide (LPS; 5 mg/kg) or underwent mid-grade cecal ligation and puncture (CLP) (Fig 1A and S1 Table), both of which are commonly used to induce systemic inflammation related to bacterial infections [17]. Up-regulated and down-regulated genes exhibited a high degree of similarity between LPS and CLP treatment (S1A Fig). Principal component analysis (PCA) of the transcriptomic profiles revealed that both LPS and CLP treatments led to clusters distinct from sham and saline controls, with the LPS treatment producing more consistent responses (Figs 1B and S1B). Gene Set Enrichment Analysis

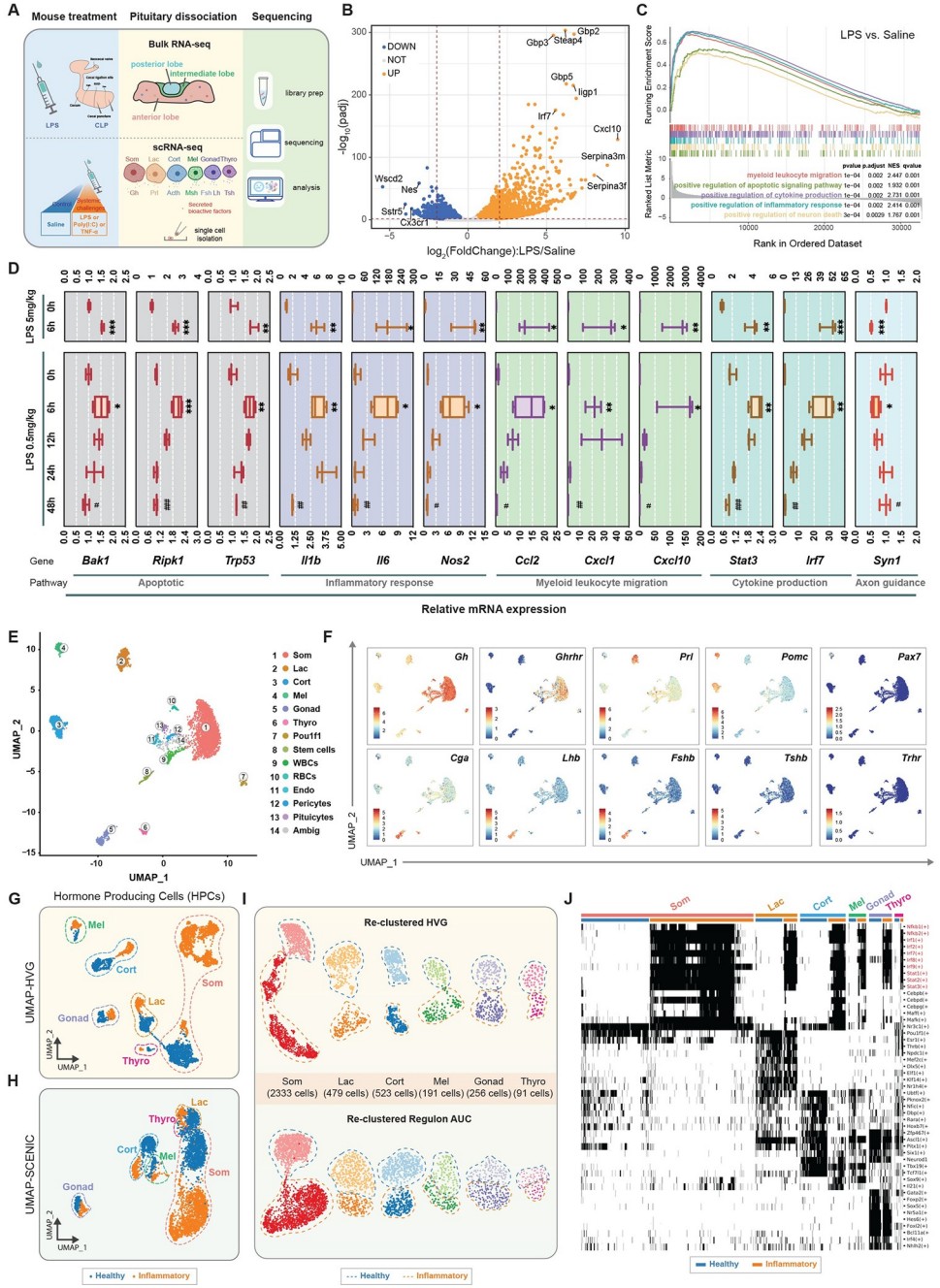

**Fig 1. scRNA-seq reveals the transcriptional landscape of the pituitary under systemic inflammation.** (**A**) Schematic diagrams of the experimental setup. The representations of syringe and liquid drop were created with BioRender.com. (**B**) Volcano plot showing the pituitary gene expression from mice subjected to 5 mg/kg LPS for 6 h and the control groups treated with saline. Genes up-regulated or down-regulated by log$_2$(FoldChange) >|0.5| are shown in red and blue, respectively. (**C**) GSEA profiles showing significant enrichment of gene sets after LPS treatment in the pituitary. The transcriptome datasets used in (**B**) were for GSEA analysis. (**D**) qPCR analysis of representative genes enriched in gene sets identified in (**C**) under 0.5 mg/kg and 5 mg/kg LPS treatments and at different time points (*n* = 3–4 mice). (**E**) UMAP plot showing pituitary cell populations from all treatment groups after Seurat v3 integration. Legends indicate cell types. (**F**) Scatterplots showing expression of canonical marker genes for HPCs, overlaid on the same embedding as in (**E**). (**G**) UMAP plot of HPCs from the saline- and LPS-treated groups showing different cell types and cell states based on HVGs. Dashed line: cell types; dot color: cell states. (**H**) UMAP plot of HPCs from the saline- and LPS-treated groups showing different cell types and cell states based on SCENIC regulon AUC scores. Dashed line: cell types; dot color: cell states. (**I**) Re-clustering of HPCs in (**G**) and (**H**). Dashed

line: cell states. (**J**) Heatmap showing the representative binarized SCENIC regulon activity in HPCs with healthy and inflammatory cell states predicted in (**I**). Columns are cells and rows are regulons. White: not activated; black: activated. All data represent mean with SD. Statistical significance was determined by two-tailed Student's *t* test, \*$p < 0.05$, \*\*$p < 0.01$, \*\*\*$p < 0.001$. The data underlying this figure can be found in S1–S3 Tables and S1 Data. GSEA, Gene Set Enrichment Analysis; Som, somatotropes; Lac, lactotropes; Cort, corticotropes; Mel, melanotropes; Gonad, gonadotropes; Thyro, thyrotropes; Pou1f1, proliferating pou1f1-positive cells; WBCs, white blood cells; RBCs, red blood cells; Endo, endothelial cells; Ambig, ambiguous; UMAP, uniform manifold approximation and projection; HVGs, highly variable genes; HPCs, hormone-producing cells; SCENIC, single-cell regulatory network inference and clustering; AUC, area under the curve; LPS, lipopolysaccharide; scRNA-seq, single-cell RNA-sequencing.

(GSEA) revealed strong up-regulation in the expression of genes involved in the signaling pathways related to apoptosis, proinflammatory response, cytokine production, and myeloid leukocyte migration (Figs 1C and S1C). qPCR tests confirmed significant increases in the expression of several representative genes, such as *Ripk1*, *Il6*, *Stat3*, and *Ccl2* that were found to be enriched in the gene sets in GSEA analysis (Fig 1D, top panel). We further explored the progression of inflammation in the pituitary by treating mice with a subseptic dose of LPS (0.5 mg/kg, i.p.) and examining gene expression using qPCR at various time points (6, 12, 24, and 48 h postinjection). We found that the pituitary gland responded dynamically to inflammatory processes, displaying intense disturbance at 6 h and gradually returning to the basal state from 12 to 48 h (Fig 1D). Taken together, these results indicate that the pituitary cells at the population level exhibit potent and dynamic responses to systemic inflammation.

To map the transcriptomic responses with cell type-specificity, we performed scRNA-seq based on the modified single-cell tagged reverse transcription sequencing (STRT-seq) method on pituitary cells from mice treated with different doses of LPS (0.5, 1, 10, and 50 mg/kg) at different time points (3, 6, 24, 48 h and 3 weeks). We also challenged mice with polyinosinic-polycytidylic acid (Poly(I:C)) and tumor necrosis factor α (TNF-α) to investigate the potential effects of viral infection and proinflammatory cytokine, respectively, to broaden our understanding of the effects of systemic inflammation in the pituitary (Fig 1A). After conducting quality control checks on gene recovery, unique molecular identifiers (UMIs), and the percentage of ribosomal and mitochondrial RNA, we retained 5,506 high-quality cells for subsequent analysis (S2A–S2C Fig and S2 Table).

Utilizing the Seurat anchor method [18], we coalesced cell populations across treatments into a common space. Employing the unsupervised Louvain algorithm [19], we delineated 14 clusters, 13 of which exhibited distinct cell identities (Fig 1E). These clusters encompassed all 6 major types of secretory HPCs, as identified by the established markers, including hormone and hormone receptor genes (*Gh* and *Ghrhr* for somatotropes; *Prl* for lactotropes; *Pomc* for corticotropes and melanotropes; *Cga*, *Lhb*, and *Fshb* for gonadotropes; *Cga* and *Tshb* for thyrotropes), as well as a transcription factor gene (*Pax7* for melanotropes) (Fig 1F). Additionally, we identified 7 auxiliary cell types by their characteristic marker expression, including white blood cells (WBCs, *C1qb*+), red blood cells (RBCs, *Hbb-bt*+), stem cells (Stem, *Sox2*+ and *Cyp2f2*+), proliferating Pou1f1 cells (Pou1f1, *Pbk*+ and *Mki67*+), endothelial cells (Endo, *Pecam1*+ and *Emcn*+), pericytes (Peri, *Col1a1*+ and *Ogn*+), and pituicytes—the pituitary glia cells (Pitui, *Col25a1*+ and *Scn7a*+) [16]. We also found a minor cluster containing a few single cells with ambiguous identity (cluster 14) (S2E Fig). Evaluating cell proportions, we observed that immune stimuli did not modify the pituitary cell type composition (S2D Fig).

We conducted more detailed analyses on HPCs, which are the principal cells responsible for hormone secretion in the pituitary [12]. To investigate the gene expression patterns of the HPCs from the saline- and LPS-treated groups, we extracted the top 2,000 highly variable genes (HVGs) (S3 Table). Using uniform manifold approximation and projection (UMAP) [20], we embedded the HPCs into an unintegrated, naïve, and low-dimensional space. In the

UMAP-HVG embedding, pituitary cells could be segregated into 6 HPCs and 2 cell states, based on variations in their molecular signatures (Fig 1G). Re-clustering with the Louvain algorithm supported that each HPC can be subdivided into binary cell states (Fig 1I, top panel), which we named "healthy" or "inflammatory" states, respectively.

To confirm that the HPCs were indeed segregated into binary states, we applied single-cell regulatory network inference and clustering (SCENIC) to analyze the data at the regulon level [21]. SCENIC robustly aggregates gene expression at the transcription factor level and yields a regulon score (AUCell) that represents the regulon activity. Consistent with the UMAP-HVG analysis, the UMAP-SCENIC embedding clustered pituitary cells according to cell types and cell states (Fig 1H), with each cell type identified as 2 state-specific clusters (Fig 1I, bottom panel). The Pou1f1-dependent populations, including somatotropes, lactotropes, and thyrotropes [13,22,23], formed indiscernible boundaries, while corticotropes and melanotropes that are Pou1f1-independent populations with some shared marker genes, such as *Pomc*, clustered closer to each other and away from the Pou1f1-dependent populations (Fig 1H). These results indicate that the SCENIC analysis accurately recapitulates the healthy and inflammatory cell states in control and immune-challenged conditions, as well as the classical lineage relationships of pituitary HPCs. More importantly, the binary activity matrix of 356 SCENIC regulons revealed that HPCs in the healthy and inflammatory states exhibited both unique and shared features (S4 Table and S2F Fig), especially those related to highly enriched members of the Irf/Stat family, across different cell types (Fig 1J).

To investigate the contribution of HPCs to the pituitary inflammatory response process, we compared gene expression changes in the pituitary tissue from the RNA-seq dataset and HPCs from the scRNA-seq dataset after LPS treatment. Correlation analysis indicated that pituitary HPCs exhibited a highly similar transcriptomic signature to that of the entire pituitary gland (S2G Fig).

To explore the impact of Poly(I:C) and TNF-α on the pituitary, we employed Seurat label transfer to apply the "healthy" or "inflammatory" state designations, as established by the saline and LPS treatment groups, to the cells in the Poly(I:C) and TNF-α treatment groups (Figs 1G and S3A). UMAP and alluvial plots for the 6 HPCs demonstrated that the HPCs in the Poly(I:C) and TNF-α treatment groups also delineated into the binary "healthy" or "inflammatory" cell states, akin to those observed in the LPS treatment (S3B–S3E Fig).

Collectively, our high-dimensional pituitary single-cell atlas indicated that, at the transcriptional level, all pituitary HPCs respond strongly to systemic inflammation. The high-quality scRNA-seq dataset allowed us to further examine the properties of inflammatory responses by pituitary cells in a cell type-specific manner.

## Corticotropes respond strongly to systemic inflammation

To investigate the distinct features and biological functions of HPCs under inflammation, we first performed differential expression analysis by calculating conserved and differentially expressed marker genes using healthy and inflammatory states as grouping factors (Fig 2A). We identified 418 conserved marker genes in HPCs that can serve as markers for identifying cell types and ensuring the quality of the sequencing data (S5 Table and Fig 2B, top panel). For example, *Car10* was a conserved marker for somatotropes across healthy and inflammatory states, and so was *Edil3* for lactotropes, *Pall a* for corticotropes, *Megf11* for melanotropes, *Ttc24* for gonadotropes, and *Pvalb* for thyrotropes (S4A–S4F Fig). Next, we identified 969 differentially expressed genes (DEGs) across all HPCs, which can be used to study the specific cellular responses to inflammation (S6 Table and Fig 2B, bottom panel). UpSet plot visualization of the conserved marker genes and DEGs across the pituitary HPCs revealed that corticotropes had

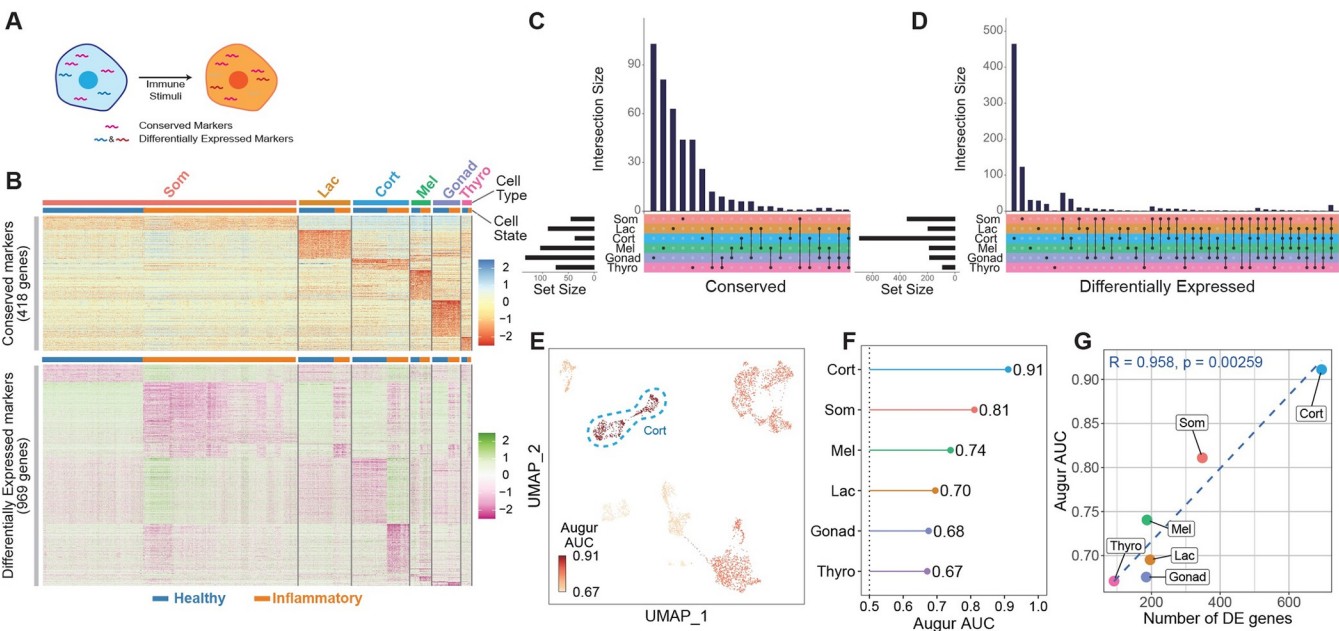

**Fig 2. Differential expression and Augur analysis show corticotropes as the most responsive HPCs under systemic inflammation.** (**A**) Schematic diagrams of the conserved and differentially expressed markers in HPCs subjected to immune stimuli. (**B**) Heatmaps showing the conserved marker genes (top panel) and DEGs (bottom panel) for each cell type under healthy or inflammatory states ($\log_2$(FoldChange) > 0.65 and $p$-value adjusted < 0.001). (**C and D**) Upset plots depicting the relationship of conserved markers (**C**) or DEGs (**D**) that are unique to or shared between cell types. Bars indicate the number of genes. Lines between cell types highlight shared conserved marker genes (**C**) or DEGs (**D**). (**E–G**) Augur analysis of HPCs in response to inflammation. Cell types with AUC scores closer to 1 responded more strongly to inflammation. (**E**) Scatter plot shows the Augur AUC intensity of individual cells. (**F**) Average Augur AUC for each major HPC cell type. (**G**) Plot of Augur AUC and the number of DEGs. The data underlying this figure can be found in S5–S8 Tables. DEG, differentially expressed gene; HPC, hormone-producing cell.

the largest number of unique DEGs among the HPCs, while the number of conserved genes did not show difference among cell types (S7 and S8 Tables and Fig 2C and 2D). Gene pathway enrichment analysis of the unique DEGs in each HPC revealed that aside from a consensus response in the inflammatory pathway, different cell types also exhibited unique responses in other pathways (S9 Table and S5 Fig). For example, somatotropes displayed a lymphocyte activation response, whereas melanotropes displayed an extrinsic apoptotic signaling response (S5G Fig). Overall, these results demonstrate that different types of HPCs exhibit distinct response features to systemic inflammation.

In agreement with the above findings, Augur AUC analysis [24] on 6 HPCs revealed that corticotropes responded most strongly to the inflammatory challenges (Fig 2E–2G). Gene Ontology (GO) analysis of corticotrope-specific DEGs revealed that genes involved in the pathways of protein modification and phosphorus metabolic process (representative genes: *Sphk1*, *Gprc5a*, *Ptpn2*, and *Rnf180*) were greatly up-regulated (S6A and S6C–S6F Fig and S10 Table), while the highly enriched pathways among the down-regulated DEGs primarily involve nucleoside triphosphate metabolic processes (representative genes: *Gamt*, *Acaa2*, *Nme3*, and *Atp1a2*) in the inflammatory state (S6B and S6G–S6J Fig and S10 Table). Taken together, corticotropes are the most perturbable cells in the pituitary during inflammation.

## Systemic inflammation modulates the secretory functions of pituitary HPCs

The pituitary is best known for its functions of secreting peptide hormones. Under immune stress conditions, the pituitary corticotropes secrete ACTH, which promotes the serum levels of corticosterone [25], suggesting an essential function in pituitary secretion during

inflammation. Consistent with this view, the transcription level of *Pomc*, a precursor polypeptide of ACTH, was up-regulated at a moderate but statistically significant level in corticotropes under inflammatory states compared to healthy states (S7A Fig). We also observed that the transcription levels of genes encoding GH, PRL, LH, and TSH underwent varying degrees of changes under inflammation (S7A Fig), thus indicating the potential effects of systemic inflammation on modulating the production of classic pituitary hormones.

We further asked whether inflammation might trigger the production of other secreting factors by the pituitary. GO analysis of 969 DEGs in pituitary HPCs revealed that many of them were enriched in the "extracellular region" and "cytokine-mediated signaling pathway" (S11 Table and Fig 3A and 3B). To validate our transcriptomic data, we performed proteomic analysis of the pituitary following LPS treatment and analyzed differentially abundant proteins using GO analysis (S12 Table and S7B Fig). The enriched proteins were also associated with the "extracellular space" and "extracellular region" GO terms (S7C Fig), which corroborated the transcriptomic results. The scRNA-seq dataset showed that the expression levels of myeloid migration-related chemokines, such as *Ccl2*, *Cxcl1*, and *Cxcl10* [26–28], were strongly up-regulated in all HPCs under systemic inflammation (S7D Fig). These results suggest that pituitary HPCs may release non-hormone bioactive substances, such as cytokines, in response to inflammatory challenges.

Beyond genes encoding secreting factors, numerous DEGs involved in the maturation of secreted proteins or the regulation of secretion processes are transcriptionally perturbed. For example, *Sult1a1*, a gene encoding sulfotransferase that is responsible for catalyzing sulfate conjugation of hormone molecules to increase their solubilities [29], displayed pronounced up-regulation in somatotropes during systemic inflammation (S8A Fig). Similarly, *Sphk1*, the gene encoding a kinase that phosphorylates sphingosine to sphingosine-1-phosphate (S1P)—a sphingolipid involved in the regulation of inflammatory responses [30], was also up-regulated (S6C Fig). By contrast, *Sstr5* and *Crhr1*, the 2 neuropeptide receptor genes for somatostatin and CRH from hypothalamic neurons to control pituitary hormone release [31,32], were markedly down-regulated (S8B–S8D Fig). Collectively, these results indicate that systemic inflammation strongly affects the secretory processes of pituitary HPCs.

## Nptx2 is a corticotrope-specific factor that modulates systemic inflammation

Next, we investigated whether systemic inflammation might lead to the release of novel pituitary-specific factors to modulate immune responses. We chose a list of 13 candidate genes encoding secreted proteins that were induced in at least 1 type of HPC during inflammation (Fig 3C). Most of these candidate genes are involved in the immune or inflammation process, such as *Fgg*, *Cartpt*, *Isg15*, *Lbp*, *Lgals9*, *Fndc4*, *Il1r2*, and *Serpina3n*, according to previous studies [33–40]. Additionally, *Nptx2*, *Cartpt*, and *Vgf* are involved in nerve system processes [34,41–43], and *Insl6* participates in the reproductive process [44].

To address whether induced expression of DEGs is specific to pituitary HPCs, we performed qPCR to examine the expression levels of the 13 candidate genes in multiple tissues, including the pituitary, all major peripheral organs, and 3 major brain areas, from saline- and LPS-treated mice. This screening identified 2 gene products, Neuronal pentraxin 2 (*Nptx2*) and Cocaine- and amphetamine-regulated transcript (*Cartpt*), that were robustly and specifically up-regulated in the pituitary gland during systemic inflammation (Fig 3D). The scRNA-seq data showed that *Nptx2* was specifically up-regulated in corticotropes, while *Cartpt* was up-regulated in corticotropes and somatotropes (Figs 3C, S8E, and S8F). CART peptide (encoded by *Cartpt*) has an established role in immunomodulation [34,45], whereas our

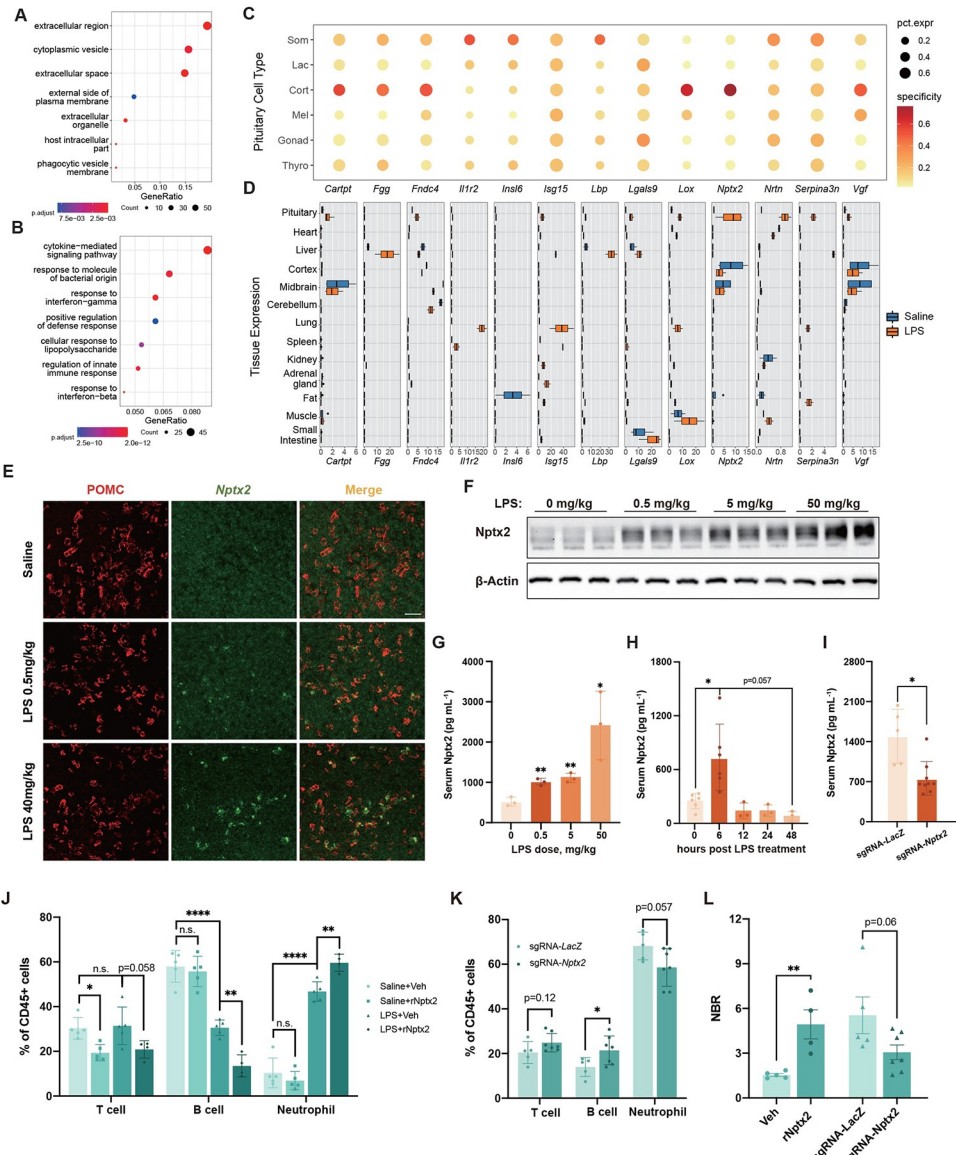

**Fig 3. Nptx2 is specifically released by corticotropes in response to systemic inflammation.** (**A** and **B**) GO analysis of DEGs in HPCs showing enriched GO terms in biological process (**A**) and cellular component (**B**) categories. (**C**) Dot plots showing the 13 candidate genes encoding secreted proteins in HPCs. Dot size and color represent expression abundance and the expression specificity of the gene, respectively. (**D**) qPCR screening of the 13 genes in multiple tissues from control and LPS-treated mice ($n$ = 2–4 mice). (**E**) Representative images showing ISH of *Nptx2* RNA (green) and IF of POMC (red) in the pituitary from mice treated with saline, 0.5 mg/kg or 40 mg/kg LPS. Scale bar, 50 μm. (**F**) Immunoblot analysis of Nptx2 in the pituitary from mice treated with saline or LPS ($n$ = 3 mice). (**G**) Serum concentration of Nptx2 from mice treated with different doses of LPS ($n$ = 3). (**H**) Serum concentration of Nptx2 from mice treated with the same dose of LPS for different durations ($n$ = 3–6 mice). (**I**) Serum concentration of Nptx2 from mice subjected to AAV-SaCas9-U6-sgRNA-*Nptx2* in the pituitary for 3 weeks, followed by LPS exposure ($n$ = 5 or 9 mice per group). (**J**) FACS analysis of T cells, B cells, and neutrophils in PBMCs from mice pretreated with rNptx2, followed by saline or LPS treatment ($n$ = 4–5 mice). (**K**) FACS analysis of T cells, B cells, and neutrophils in PBMCs from mice that received the same treatments as shown in (**I**) ($n$ = 5–7 mice). (**L**) Analysis of the NBR from (**J**) LPS +vehicle and LPS+rNptx2 groups, and (**K**) sgRNA-*LacZ* and sgRNA-*Nptx2* groups. The injection dose of LPS was 0.5 mg/kg in (**D, H, I, J, and K**). The time duration of LPS was 6 h in (**D, E, F, G, I, J, and K**). All data represent mean with SD. Statistical significance was determined by two-tailed Student's $t$ test (**G, H, I, K, and L**) or by one-way ANOVA with Tukey's multiple-comparisons test (**J**), *$p$ < 0.05, **$p$ < 0.01, ****$p$ < 0.0001. The data underlying this figure can be found in S11 Table and S1 Data. The original blot for this blot can be found in S1 Raw Images. DEG, differentially expressed gene; GO, Gene Ontology; ISH, in situ hybridization; IF, immunofluorescence; POMC, proopiomelanocortin; HPCs, hormone-producing cells; LPS, lipopolysaccharide; NBR, neutrophil-to-B cell ratio; PBMC, peripheral blood mononuclear cell.

current understandings of Nptx2 functions are limited to synapse formation in normal brain functions and neurodegenerative diseases [41,46]. We thus investigated the potential roles of Nptx2 in immunomodulation.

In the pituitary, the gene expression level of *Nptx2* showed a 20-fold increase at 6 h after LPS treatment and then gradually returned to basal levels 12 to 48 h later (S8G Fig). RNA in situ hybridization (ISH) analysis confirmed that intraperitoneal administration of LPS dose-dependently increased the percentage of *Nptx2*+ corticotropes (Figs 3E and S8H), a result that was further confirmed by western blotting analysis of Nptx2 protein (Fig 3F).

Consistently, serum Nptx2 protein levels displayed a dose-dependent increase 6 h after LPS treatment (Fig 3G), followed by a transient decline, with serum Nptx2 returning to basal levels 12 h post LPS treatment (Fig 3H). Through adeno-associated virus (AAV)-mediated CRISPR/Cas9 genome editing, we disrupted Nptx2 expression. Stereotaxic microinjection of AAV vectors into pituitary substantially reduced pituitary Nptx2 expression and serum Nptx2 levels during LPS treatment (Figs 3I and S8I). In unison, these findings indicate the release of Nptx2 from the pituitary into the circulatory system in response to inflammatory reactions.

LPS treatment induces strong inflammatory responses, resulting in the production of a diverse range of proinflammatory cytokines. To elucidate the regulatory mechanism of *Nptx2* up-regulation within the pituitary, we investigated whether this effect was directly mediated by LPS or through the actions of proinflammatory cytokines. For this purpose, we employed mouse AtT-20 cells, a well-established model for ACTH-secreting corticotropes in the pituitary [47,48]. Treating AtT-20 cells with serum obtained from LPS-challenged mice or the proinflammatory cytokine IL-6, instead of LPS, IL-1α, IL-1β, or TNF-α, significantly up-regulated the expression levels of *Nptx2* (S8J and S8K Fig). These results suggest that LPS-induced *Nptx2* expression is mediated by LPS-triggered proinflammatory cytokines, such as IL-6.

A recent study highlights that the interaction between Nptx2 and complement C1q influences microglia activity within the brain [49]. We delved into the potential functional impact of Nptx2 on the peripheral immune system, specifically focusing on how it might affect the blood neutrophil-to-lymphocyte ratio (NLR), which is a marker of peripheral inflammation and immune system homeostasis [50,51]. We isolated peripheral blood mononuclear cells (PBMCs) from saline- and LPS-treated mice using CD45-based cell sorting and calculated the percentage of neutrophils and 2 major types of lymphocytes (T cells and B cells) within the PBMC population. We found that the intravenous injection of recombinant Nptx2 resulted in a similar reduction in the percentage of T cells in saline- and LPS-treated mice. Consistent with previous observations [52], LPS treatment significantly reduced the percentage of B cells and increased the percentage of neutrophils. More importantly, the intravenous injection of Nptx2 further enhanced this effect, leading to an increase in the neutrophil-to-B cell ratio (NBR) (Fig 3J and 3L).

To test whether Nptx2 in the pituitary contributes to immune modulation, we disrupt Nptx2 expression in the pituitary (S8I Fig), which resulted in the abolishment of the effect of LPS on reducing B cells and increasing neutrophils (Fig 3K and 3L). This suggests that the pituitary modulation of NBR depends on Nptx2. Taken together, these results indicate that Nptx2 acts as a novel pituitary hormone and may have a functional role in balancing systemic inflammation and immunity [53].

## Crosstalks between the pituitary and immune cells during systemic inflammation

Our findings thus far suggest that pituitary HPCs respond profoundly to systemic inflammation and may release cytokines as well as other novel immunomodulators to affect the immune

system (Figs 3B and S7D). To probe the potential bidirectional endocrine communications between the pituitary and the immune system, we performed cell–cell communication analysis with CellChat [54] using our pituitary scRNA-seq dataset and a previously published scRNA-seq dataset on the spleen [55]. Systemic inflammation was associated with more and overall stronger interactions within and between the pituitary and spleen populations (Fig 4A). More

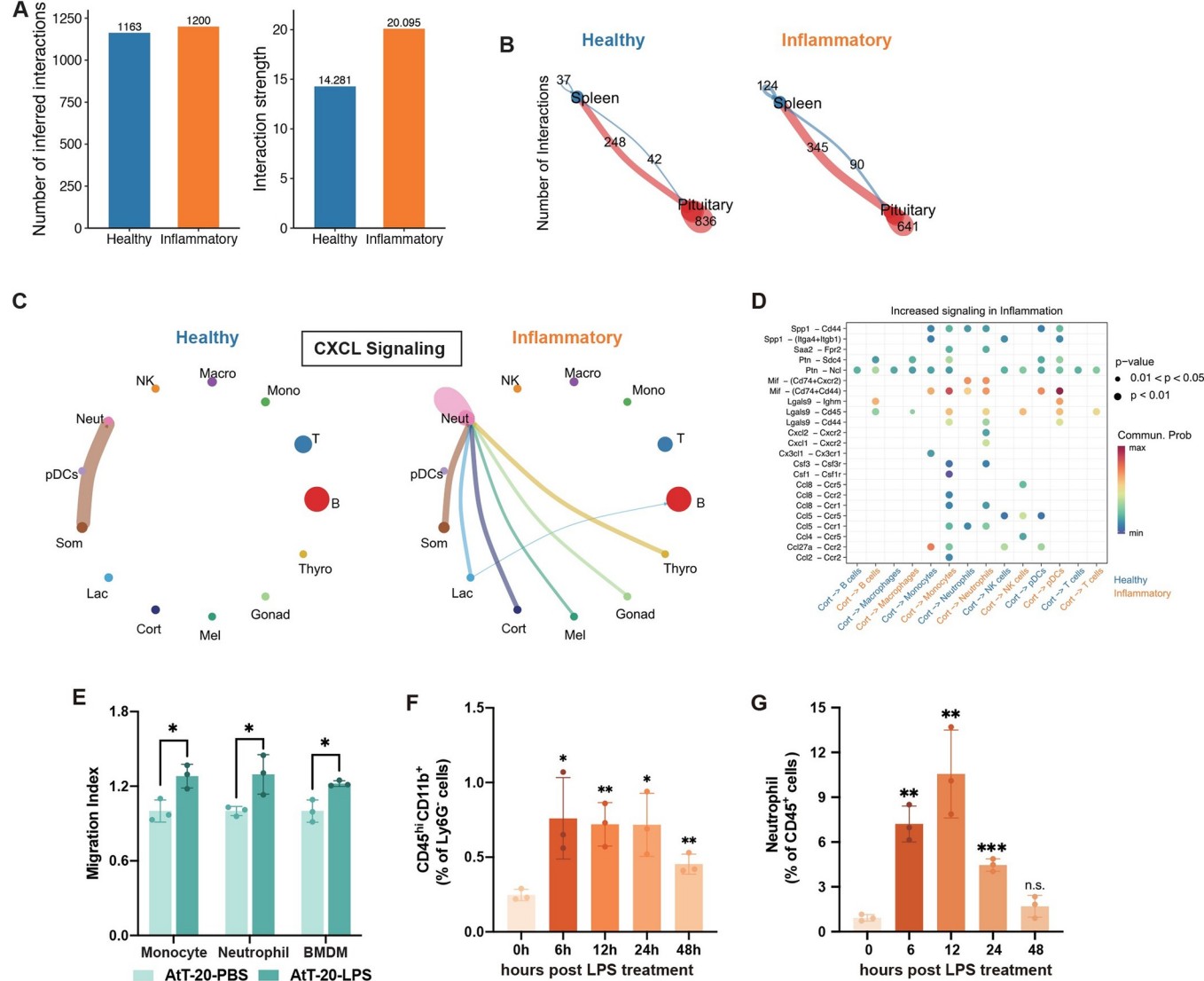

**Fig 4. Cell–cell communications between the pituitary and the immune cells under healthy or inflammatory states.** (**A**) Bar graph showing the total number (left) and strength (right) of possible interactions between pituitary cells and spleen cells under healthy or inflammatory state by CellChat analysis. (**B**) Circle plots showing the number of inferred interactions within and between pituitary and spleen populations. Red line: the interactions from the pituitary to the spleen; blue line: the interactions from the spleen to the pituitary. (**C**) Circle plots showing the number of statistically significant interactions for the CXCL pathways between HPCs in the pituitary and immune cells in the spleen. Each color represents 1 cell type; edges connecting circles represent significant intercellular signaling inferred between those cell types. (**D**) Inferred interactions from corticotropes to splenic immune cell populations under healthy or inflammatory states. The dot color and size represent the calculated communication probability and p-values; p-values are computed from one-sided permutation test. (**E**) Transwell index of monocyte, neutrophil, and macrophage recruited by PBS- or LPS-treated AtT-20 cells for 6 h ($n = 3$ replicates). (**F** and **G**) Flow cytometry analysis of macrophages (**F**) and neutrophils (**G**) in the pituitary from mice treated with saline or LPS (0.5 mg/kg; $n = 3$ mice). All data represent mean with SD. Statistical significance was determined by two-tailed Student's $t$ test, $*p < 0.05$, $**p < 0.01$, $***p < 0.001$. The data underlying this figure can be found in S1 Data. HPC, hormone-producing cell; LPS, lipopolysaccharide; NK, natural killer cell; Macro, macrophage; Neut, neutrophil; pDCs, plasmacytoid dendritic cells; Mono, monocyte; T, T cell; B, B cell; Som, somatotropes; Lac, lactotropes; Cort, corticotropes; Mel, melanotropes; Gonad, gonadotropes; Thyro, thyrotropes.

specifically, during systemic inflammation putative signaling increased between the pituitary and spleen populations and within the spleen, whereas it decreased within the pituitary (Fig 4B). These data suggest that inflammation increases the probability of endocrine signaling between the pituitary and the spleen.

Next, we explored the communication between pituitary HPCs and spleen immune cells in a cell type- and signaling pathway-specific manner. Systemic inflammation drastically increased the information flow in CXCL signaling between immune cells, especially neutrophils, and all major types of HPCs (Fig 4C). Corticotropes, the prominent responders to inflammation in HPCs, mediated the enhanced connections in CCL and CXCL signaling interactions with many immune cells, especially monocytes and neutrophils (Fig 4D). These results suggest that systemic inflammation promotes the cell–cell signaling between the pituitary cells and immune cell populations in a signaling pathway-specific manner.

Given the convenience of studying pituitary–immune cell–cell communications in vitro, we chose to functionally validate the communications using AtT-20 cells. Treating AtT-20 with the serum from LPS-challenged mice or with one of 5 inflammatory stimuli (LPS, IL-1α, IL-1β, TNF-α, and IL-6) significantly up-regulated the expression levels of multiple chemokine-encoding genes (S9A and S9B Fig), demonstrating that inflammatory cytokines could directly modulate the gene expression of corticotropes. Using Transwell migration assay, we found that after being treated with LPS, AtT-20 cells attracted neutrophils, monocytes, and bone marrow-derived macrophages (BMDMs) (Fig 4E), indicating that immune cells can be recruited by inflammatory corticotropes during inflammation in vitro.

At the in vivo level, fluorescence-activated cell sorting (FACS) results showed that the number of macrophages ($CD45^{hi}CD11b^+$) [56] increased in the pituitary and returned to normal levels 48 h after LPS treatment, which were consistent with the macrophage marker gene expression level (Figs 4F, S9C–S9E, and S9H). The number of $CD206^+$ M2-like macrophages, a population responsible for tissue repair [57,58], increased 24 h after LPS treatment and peaked at 48 h (S9I–S9L Fig). We also observed dynamic changes in the number of neutrophils in the pituitary (Figs 4G, S9C, S9F, and S9G). Taken together, these results corroborated the findings of the CellChat analysis and suggested that the pituitary is modulated by systemic inflammation to recruit immune cells into the pituitary likely through enhanced chemokine signaling.

## Macrophages promote ACTH release under inflammation

Finally, we investigated the potential effects of immune cell activity on pituitary cells during inflammation. Since we only observed neutrophils adhering to the blood vessel walls and did not observe infiltration into the pituitary parenchyma (S9G Fig), we focused on the function of macrophages, which are located in the parenchyma (S9E Fig) and possess well-established methods of manipulation. When co-cultured with LPS-pretreated BMDMs, AtT-20 increased the expression of *Pomc* and *Pcsk1* (Fig 5A), which are involved in the maturation of the ACTH peptide [59], and elevated ACTH concentration in the supernatant of the co-culture medium (Fig 5B), indicating that macrophages can affect hormone production in pituitary cells. Next, we treated mice with clodronate-containing liposomes to deplete macrophages [60] via stereotaxic microinjection into the pituitary (Fig 5C). After macrophage depletion, the concentrations of ACTH and corticosterone decreased in the serum of LPS-treated mice, whereas liposome-clodronate treatment did not affect baseline ACTH production (Fig 5D and 5E). This suggests that macrophages within the pituitary promoted the release of ACTH during inflammation. As CCL2 governs macrophage migration during inflammation [26], we investigated the in vivo impact of pituitary CCL2 overexpression. AAV-mediated CCL2 expression

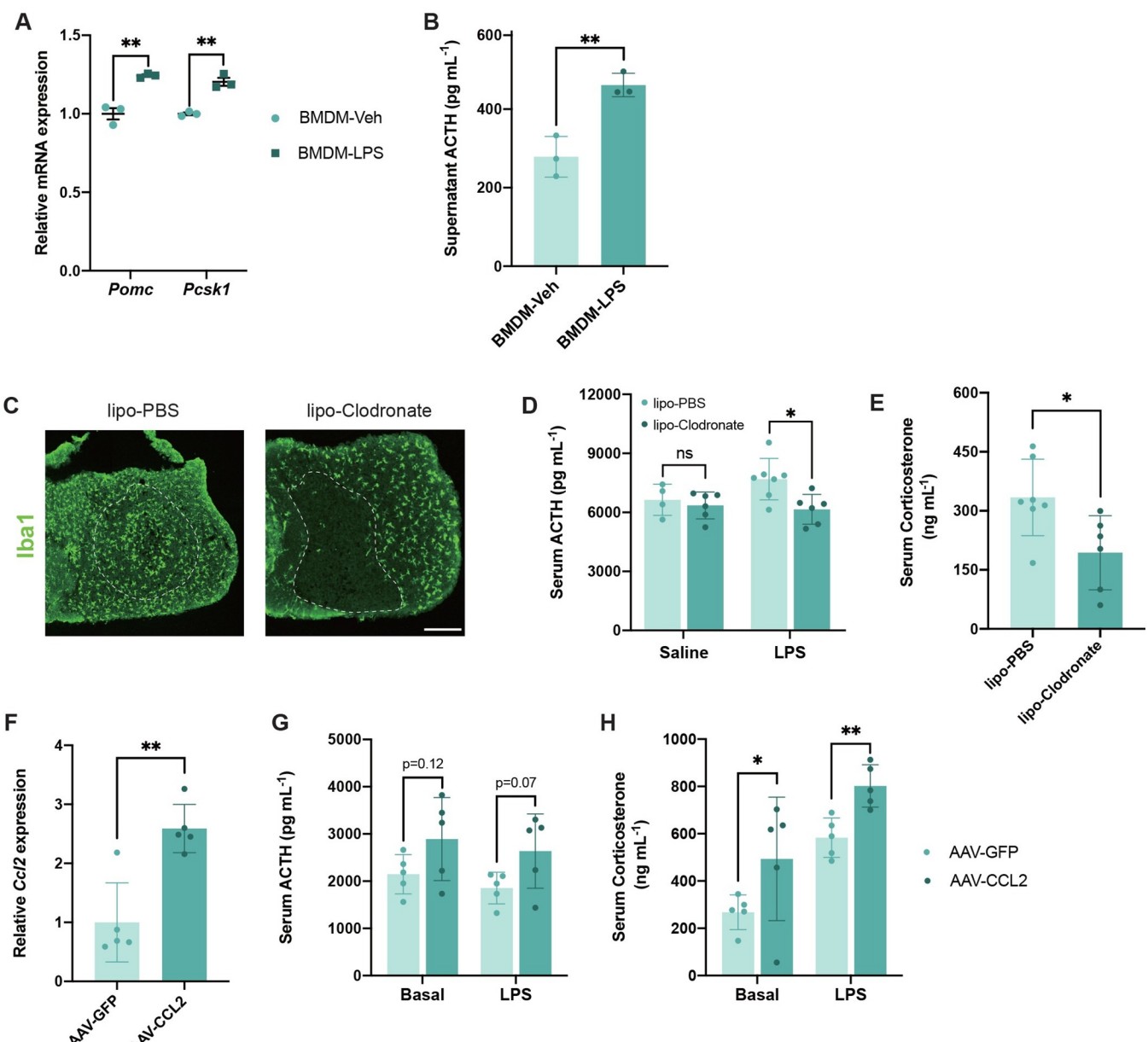

**Fig 5. Macrophages in the pituitary facilitate ACTH secretion under inflammation.** (**A**) Gene expression of *Pomc* and *Pcsk1* in AtT-20 cells after co-culture with vehicle (Veh) or LPS-treated BMDMs for 6 h (*n* = 3 replicates). (**B**) ACTH concentration in the supernatant from AtT-20 cells after co-culture with Veh or LPS-treated BMDMs for 6 h (*n* = 3 replicates). (**C**) Representative images showing Iba1 (the marker of macrophage) expression in the pituitary from mice that received liposome-PBS (left) or liposome-Clodronate (right) directly to the pituitary for 24 h. Scale bar, 100 μm. (**D**) Serum concentrations of ACTH in saline- or LPS-treated (0.5 mg/kg LPS for 6 h) mice pretreated with liposome-PBS or liposome-Clodronate (*n* = 4–7 mice). (**E**) Serum concentrations of corticosterone in LPS-treated (0.5 mg/kg LPS for 6 h) mice pretreated with liposome-PBS or liposome-Clodronate (*n* = 6–7 mice). (**F**) qPCR validation of the *Ccl2* transcript in mice subjected to AAV-GFP or CCL2 in the pituitary for 3 weeks (*n* = 5 mice). (**G** and **H**) Serum concentrations of ACTH (**G**) and corticosterone (**H**) before and after LPS treatment (0.5 mg/kg for 6 h) in pituitary GFP- or CCL2-overexpressing mice (*n* = 5 mice). All data represent mean with SD. Statistical significance was determined by two-tailed Student's *t* test, *$p < 0.05$, **$p < 0.01$. The data underlying this figure can be found in S1 Data. AAV, adeno-associated virus; BMDM, bone marrow-derived macrophage; LPS, lipopolysaccharide.

in the pituitary elevated serum concentrations of ACTH and corticosterone under both normal and immune-challenged conditions (Fig 5F–5H). Taken together, these results indicate that macrophages in the pituitary promote ACTH release and further support functional interactions between pituitary cells and immune cells under systemic inflammation.

## Discussion

The HPA axis plays an essential role in the regulation of body homeostasis. As a major component of the HPA axis, the pituitary has been extensively studied in the context of stress responses and immune challenges. Previous studies have focused on corticotropes in the pituitary, which receive CRH signals from the PVN in the hypothalamus and release the peptide hormone ACTH. The release of ACTH leads to an increase in the serum concentrations of glucocorticoid, which activates the glucocorticoid receptor to control inflammatory gene expression and ultimately suppresses inflammation. This mechanism has served as the basis for the treatment of various clinical diseases, such as acute inflammatory syndromes related to cytokine storm following SARS-CoV-2 infection [61]. However, it remains unknown how systematic inflammation affects the transcriptomic profiles of pituitary cells. Here, we used scRNA-seq and other omics techniques to profile the transcriptional landscape of pituitary populations under healthy and inflammatory conditions. Our data reveal wide-ranging and cell type-specific responses of pituitary HPCs to systemic inflammation and have several functional implications on the pituitary regulations of body physiology.

Firstly, nearly all pituitary cell types respond robustly and strongly to systemic immune challenges at the transcriptional level, with corticotropes responding most potently. Therefore, immune challenges not only impact the gene expression in corticotropes but also other HPCs that release several fundamentally important hormones, such as FSH and LH regulating reproduction, GH regulating body development and shape, PRL stimulating milk production, and TSH regulating body metabolism [1]. Previous studies have shown that LPS could affect serum levels of ACTH, GH, LH, PRL, and TSH in animals and humans [62–65]. Given that the hormone levels are regulated by multiple factors, including some outside of the pituitary, it had been unclear whether inflammation modulates the transcriptomic activity of pituitary HPCs. Our scRNA-seq analyses reveal that inflammatory stress has a profound impact on the gene network in all HPCs, including those associated with peptide processing and release. Our results thus indicate that inflammation modulates the transcriptomic profiles of pituitary secretory cells to regulate the expression, processing, and release of pituitary hormones.

Secondly, we find that in the context of systemic inflammation, pituitary HPCs may release many bioactive factors in addition to classic hormones. Inflammatory challenges strongly increase the expression of genes encoding chemokines, including *Ccl2*, *Cxcl1*, and *Cxcl10*, and non-chemokines, including *Nptx2*, *Cartpt*, and *Insl6*, thus suggesting the possibility of non-classic pituitary hormones during inflammation. Many of these factors are also expressed in other tissues. For example, immune cells respond to inflammation by releasing chemokines [66], whereas Insl6 is considered an anti-inflammatory myokine [67]. Our screening of secretory proteins across major tissues revealed pituitary-specific expression of *Cartpt* and *Nptx2* during inflammation. In particular, we find that Nptx2, which has been suggested to play various roles in the brain [42,68,69], is specifically released from corticotropes under inflammatory conditions. We find that Nptx2 from the pituitary acts as a key factor of modulating the neutrophil-lymphocyte ratio in the peripheral blood, a well-established marker of disturbed immune activity during inflammation [50,51]. These findings suggest that immune perturbations induce the expression of secretory factors in the pituitary to regulate immune homeostasis. Given the findings of several novel factors and the lack of a clear understanding of their signaling mechanisms thus far, our data suggest the need of investigating the functions of the novel secretory factors from the pituitary and exploring their detailed mechanism of action during systemic inflammation.

Thirdly, our analysis of cell–cell communications reveals complex interactions between the pituitary and peripheral immune populations under healthy and inflammatory conditions.

Cytokine signaling pathways, such as CCL2, CXCL1, and CXCL10 interactions, are up-regulated under systemic inflammation. FACS and Transwell assays reveal that pituitary–immune interactions promote the migration of immune cells, including macrophages and neutrophils, into the pituitary. Serum ACTH and corticosterone levels are reduced by ablating macrophages in the pituitary and increased by overexpressing CCL2 in pituitary cells, which suggests that parenchymal immune cell populations target pituitary cells to modulate their hormone-releasing functions. The classic HPA axis emphasizes the role of hypothalamic neuron activity in controlling hormonal release from the pituitary. Our results demonstrate that, in parallel to the hypothalamic inputs, local and distal interactions with the immune system may also have strong effects on pituitary functions.

In summary, this study reveals extensive effects of inflammatory stress on pituitary gene expression and functions. We discovered that systemic inflammation profoundly changes the gene expression profile of all pituitary HPCs, increases the expression of many secretory factors in addition to classic pituitary hormones, and forms close interactions between pituitary cells and immune cells. These findings enhance our understanding of how the pituitary may mediate the widespread effects of immune challenges on body physiology. The discovery of novel pituitary factor, such as Nptx2, suggests their use as inflammation markers and immunomodulators. Finally, our dataset shall provide a platform for mining additional pituitary factors and pituitary–immune interactions for the control of inflammation.

## Materials and methods

### Mice

Animal care and use were approved by the Animal Care and Use Committee of the National Institute of Biological Sciences, Beijing, China (Approval ID: NIBS2022M0036), in accordance with the Regulations for the Administration of Affairs Concerning Experimental Animals of China. Male C57BL/6 wild-type mice (8 to 12 weeks old) were obtained from the Beijing Vital River Laboratory Animal Technology. Mice were on a 12-h on, 12-h off-light cycle (lights off at 8 AM and on at 8 PM).

### Cell lines

HEK 293T cells (CRL-3216, ATCC) were cultured in DMEM supplemented with 10% fetal bovine serum (FBS) (0099–141, Gibco) and 1% penicillin-streptomycin (P/S) (15140–122, Gibco). AtT-20 cells (CCL-89, ATCC) were cultured in F-12K medium supplemented with 15% horse serum (HS) (26050088, Thermo Fisher Scientific), 2.5% FBS, and 1% P/S. Cells were maintained at 37°C in a humidified atmosphere with 5% $CO_2$.

### Constructs

Full-length mouse *Ccl2* cDNA was reverse transcribed from RNA of mouse pituitary and subcloned into pAAV-CAG-3×Tag vector. The 3 sgRNAs targeting mouse *Nptx2* were designed using the web tool Chopchop (https://chopchop.cbu.uib.no) and subsequently synthesized and cloned into the pX601 plasmid [70]. Primer sequences are summarized in S13 Table.

### AAV packaging

The AAV vectors were packaged as previously described [71]. The AAV serotype used in this study was AAV-DJ. HEK 293T cells were co-transfected with AAV vectors and AAV helper plasmids. The cells were harvested 72 h after transfection. The virus was purified using cesium

chloride density-gradient ultracentrifugation and dialyzed into Dulbecco's phosphate-buffered saline (DPBS) solution. The viral titer was determined using RT-qPCR.

### CLP

The mice were deeply anesthetized with 250 mg/kg of tribromoethanol via i.p. injection. The abdominal fur was carefully removed using a razor blade to expose the abdominal skin. The mice were positioned on a clean surgical operating table, and a clean surgical blade was used to make a midline incision in the abdominal skin, creating a 1 to 2 cm opening. The peritoneum was meticulously dissected to expose the cecum and the connected intestine. Using a medical suture, a ligature was applied approximately halfway up from the bottom of the cecum. Next, the cecum was punctured using a 21G needle, creating 2 punctures, and gently squeezed to expel a small amount of feces. Subsequently, the cecum and the adjacent intestine were placed back into the mouse's abdominal cavity, and the peritoneum was sutured using a medical suture. Used 7 mm mouse wound clips to secure the abdominal skin. For the control group mice, the ligation and puncture steps were omitted.

### Systemic immune stimulation

For medium-dose LPS treatment, mice were i.p. injected with 5 mg/kg LPS (L4130, Sigma-Aldrich). For bulk RNA sequencing experiment, mice were killed 6 h after the induction of inflammation (which corresponds to the peak time of Clinical Severity Score [17]). For scRNA-seq experiments, mice were exposed to different immune challenges. For LPS treatment, mice were i.p. injected with a lethal dose of 50 mg/kg LPS, a septic dose of 10 mg/kg, or subseptic doses of 0.5- and 1 mg/kg LPS [72]. For acute LPS treatment, pituitaries were collected at 3, 6, 24, or 48 h after LPS treatment. To study long-term effects, pituitaries were also collected at 3, 4, and 5 weeks after LPS treatment. For Poly(I:C) treatment, mice were i.p. injected with 10 or 20 mg/kg Poly(I:C) (P1530, Sigma-Aldrich) [73], and the pituitaries were collected at 3 or 6 h or 3, 4, and 5 weeks after inflammation onset. For TNF-α treatment, mice received 0.5 mg/kg TNF-α (50349-MNAE, Sino Biological), and the pituitaries were collected at 6 h after treatment.

### Bulk RNA sequencing and bioinformatics analysis

Total RNA was extracted from the pituitary using TRIzol (15596018, Thermo Fisher Scientific) and subjected to single-end 75 bp high-throughput sequencing on an Illumina platform. RNA-seq reads were mapped to the mouse reference genome (mm10) using HISAT2 (v2.1.0, https://daehwankimlab.github.io/hisat2/). The SAM output of each HISAT2 run was sorted and converted into BAM using SAMtools (https://github.com/samtools/samtools/releases/). StringTie (http://ccb.jhu.edu/software/stringtie/; V1.3.5) was used to assemble and quantify read alignments obtained in the previous step. The statistical significance of the differential expression data was determined using DESeq2 (https://bioconductor.org/packages/release/bioc/html/DESeq2.html; V2.11.40.6). DEGs were subjected to GO (http://geneontology.org) and KEGG (https://www.genome.jp/kegg/) pathway enrichment analyses using the clusterProfiler R package (https://guangchuangyu.github.io/software/clusterProfiler/), with a $p$-value adjusted $< 0.05$ and $\log_2(\text{FoldChange}) > 0.5$.

### RT−qPCR analysis

Tissues or cells were collected at predetermined endpoints. Total RNA was isolated using the TRIzol reagent. Reverse transcription was performed using 5× All-In-One qPCR SuperMix

(AE341-02, Transgen) following the manufacturer's instructions. Differential expression of the genes of interest was quantified by RT-qPCR using 2× Taq Pro Universal SYBR qPCR Master Mix (LIN B1260LBB, Vazyme) on a CFX384 Real-Time System (Bio-Rad). The expression of target genes in all samples was calculated using the $2^{(-\Delta\Delta CT)}$ method and normalized to the *Gapdh* housekeeping gene. Primer sequences are summarized in S13 Table.

## Pituitary single-cell dissociation

Mice were deeply anesthetized and intracardially perfused with a pre-cold saline solution. The pituitaries were dissected, washed once with DPBS, and then digested with 1 mg/mL collagenase type II (C5138, Sigma-Aldrich) and type IV (C6885, Sigma-Aldrich) enzymes in Hank's Balanced Salt Solution (HBSS) (pH 7.2 to 7.4, with $Ca^{2+}$ and $Mg^{2+}$) at room temperature for 25 min. DMEM supplemented with 10% FBS was added to stop the digestion, and the cells were filtered through a 40 μm nylon mesh. The cells were washed once with DPBS and resuspended in DPBS supplemented with 1% bovine serum albumin (BSA) (37525, Thermo Scientific) to obtain a single-cell suspension.

## scRNA-seq library construction and sequencing

Single pituitary cells were randomly selected under a stereo microscope by mouth pipetting in DPBS supplemented with 1% BSA. The scRNA-seq library was constructed using a modified STRT-seq2 protocol as previously described [13,71]. Briefly, cells were lysed, and the mRNA was reverse transcribed into cDNA with an 8-nt cell barcode and unique molecular identifier (UMI) sequences in a cell lysis buffer. After cDNA pre-amplification, cells with different barcodes were pooled and labeled with a biotin modification index sequence at the 3′ end of the cDNA by 4 cycles of PCR. cDNA was broken into approximately 300 bp fragments randomly by sonication (Covaris). The 3′ end of the cDNA fragments was captured using Dynabeads MyOne Streptavidin C1 (65002, Invitrogen). Library construction was performed using a KAPA Hyper Prep Kit (KK8504, Kapa Biosystems). Finally, the cleaned libraries were sequenced on an Illumina platform with a 150 bp pair-end. Primer sequences are summarized in S13 Table.

## RNA ISH

The protocol for in situ RNA hybridization has been previously described with minor differences [74]. Mice were deeply anesthetized with an overdose of pentobarbital and perfused intracardially with diethylpyrocarbonate (DEPC)-saline, followed by paraformaldehyde (PFA, 4% wt/vol in DEPC-PBS). The pituitaries were isolated, postfixed for 4 h at room temperature, and dehydrated in 30% sucrose solution overnight. The fixed pituitary was embedded in an OTC medium (Leica) and cryosectioned to produce 30-μm-thick sections on a cryostat microtome (CM1950, Leica). The sections were rinsed with DEPC-PBS and then permeabilized with DEPC-PTW (DEPC-PBS, 0.1% Tween 20) and 0.5% Triton in 2× SSC. After acetylation, the sections were incubated with prehybridization buffer (50% formamide, 5× SSC, 5 mM EDTA (pH 8.0), 0.1% Tween20, and 0.1% CHAPS) for 2 h at 65°C. Next, the sections were hybridized with digoxigenin-labeled antisense cRNA probe for *Nptx2* or *Crhr1* for 20 h at 65°C in hybridization buffer (50% formamide, 5× SSC, 5 mM EDTA (pH 8.0), 0.1% Tween 20, 0.1% CHAPS, 0.3 mg/mL transfer RNA, 1× Denhardt's solution, and 1 μg/μL heparin). After electrophoresis to remove the unbound probe, the pituitary sections were incubated with anti-digoxigenin-POD (1:500 dilution, 11207733910, Roche) at 4°C for 36 h. The probe was detected using TSA Plus Cyanine 5 (1:100 dilution, NEL745001KT, PerkinElmer), and corticotropes were detected using a primary antibody, anti-POMC (1:1,000 dilution, H-029030, Phoenix Pharmaceuticals)

at 4˚C overnight. The samples were washed 3 times in PBS and incubated with fluorescent secondary antibodies (Goat anti-Rabbit Alexa Fluor 488, 1:500 dilution, 111-545-144, Jackson ImmunoResearch Labs) at room temperature for 2 h. The sections were then washed thrice with PBST (PBS, 0.3% Triton X-100). Pituitary sections were imaged using the Nikon A1 confocal microscope with a ×20 objective and processed using the QuPath software. Primer sequences are summarized in S13 Table.

## Immunofluorescence

Mice were deeply anesthetized with an overdose of pentobarbital and intracardially perfused with a pre-cold saline solution, followed by 4% PFA (in PBS). Then, the pituitaries were isolated. After being postfixed and dehydrated as mentioned in the RNA ISH procedure, the pituitary sections were prepared using a cryostat microtome. The sections were permeabilized with PBST and blocked in 3% BSA in PBST at room temperature for 1 h. Subsequently, the sections were incubated overnight at 4˚C with primary antibodies (anti-Iba1, 1:2,000 dilution, 019–19741, Wako; anti-Ly6G (FITC Conjugate), 1:400 dilution, 88876, Cell Signaling; anti-CD31, 2.5 μg/mL, AF3628, R&D Systems). Following that, the samples were washed 3 times in PBS and incubated with fluorescent secondary antibodies (Goat anti-rabbit-AF647, 1:500, 111-605-144, Jackson ImmunoResearch; Donkey anti-goat-AF647, 1:500, 705-605-003, Jackson ImmunoResearch) at room temperature for 2 h. Pituitary sections were imaged using the Olympus VS120 virtual microscopy slide scanning system with a ×10 objective and processed using the QuPath software.

## Protein sample preparation for LC-MS/MS analysis

Mice were i.p. injected with saline or 50 mg/kg LPS, and the pituitary was collected at 6 h after LPS treatment. Ice-cold lysis buffer (500 μL 8 M Urea with 1× protease inhibitors cocktail) was used to extract protein from the indicated pituitary, and the tissue was manually homogenized with Dounce grinder and ultrasonication, then lysates were centrifuged at 16,000g for 10 min at 4˚C. The supernatant was collected, and the total protein was determined by the BCA assay kit (Solarbio, PC0020). Protein was reduced in 5 mM dithiothreitol (DTT, Sigma-Aldrich) at 37˚C for 1 h, followed by alkylation with 20 mM iodoacetamide (Sigma-Aldrich) in dark for 1 h, then added DTT again for another 15 min. Added 8 M urea to the sample up to 1 mL. After centrifugation, transferred the supernatant to a 10 KDa molecular weight cut-off (MWCO) filter tube (Millipore) and centrifuged for 10 min, then the concentrated protein in upper reservoir was washed once with 8 M urea and once with 50 mM $NH_4HCO_3$. Resuspended the sample with 200 μL 100 mM $NH_4HCO_3$ and added 100 μL 100 μg/mL sequencing-grade trypsin (Promega), then digested at 37˚C for 12 h. After digestion, centrifuged at 14,000g for 10 min and collected the solution and dried in a vacuum centrifuge. Samples were resuspended in 100 μL of 0.1% formic acid (FA) before desalted with C18 stationary phase (Omix, Agilent). The samples dried down and resuspended in 5 μL of 0.01% FA before LC-MS/MS analysis.

## LC-MS/MS analysis

The LC-MS/MS experiments were performed on Fusion03 (Thermo Scientific Orbitrap Fusion Lumos Tribrid Mass Spectrometer). The MS raw data were analyzed by PEAKS Studio v8.5 protein database containing 4,136 entries. Trypsin was set as an enzyme. The protein expression data were filtered to have at least 70% valid abundance values in the same treatment group. Log2(FoldChange) >1 or <-1.

## BMDM primary culture

BMDM isolation and culture were performed as previously described [74]. Briefly, bone marrow was isolated from the femurs and tibias of 8- to 12-week-old male mice. The macrophage precursors purified by a Ficoll-Paque PLUS Media (17-1440-02, GE Healthcare) gradient were differentiated in DMEM supplemented with 10% FBS, 20 nM macrophage colony-stimulating factor (M-CSF) (315–02, PeproTech) and 1% P/S on bacteriological Petri dishes at 37˚C in a humidified 5% $CO_2$ incubator. After 7 to 8 days, differentiated BMDMs were digested with 5 mM EDTA for 5 min and then plated in 6-well cell culture plates or 24-well Transwell plates in DMEM supplemented with 10% FBS and 1% P/S for further experiments.

## Co-culture

BMDMs were stimulated with or without 100 ng/mL LPS for 6 h. After washing with DPBS 3 times, AtT-20 cells were added to the BMDMs and co-cultured for another 6 h. Suspended AtT-20 cells were collected for RNA extraction, and the co-culture medium was collected for ACTH detection using an ELISA kit (CSB-E06874m, Cusabio Biotech Co.) after centrifugation.

## Inflamed serum treatment

Mice were i.p. injected with saline or 0.5 mg/kg LPS for 6 h. Control or inflamed serum was collected from mouse blood. AtT-20 cells were treated with 25 μL serum per mL cell medium for 6 h.

## Neutrophils and monocytes isolation

Bone marrow was isolated from the femurs and tibias of 8- to 12-week-old male mice. The bone marrow cell suspension was then overlaid on top of the Histopaque-1077 (10771, Sigma) and Histopaque-1119 (11191, Sigma) double gradient. After centrifuge at 2,000 rpm for 30 min at room temperature, neutrophils were collected at the interface of the Histopaque-1119 and Histopaque-1077 layers, and monocytes were found in the upper part of Histopaque-1077. Both the neutrophils and monocytes were ready for use after washing with DPBS.

## Transwell migration assay

Transwell migration assay was performed using 6.5 mm transwell plates with 5 μm pore inserts (CLS3421-48EA, Corning). AtT-20 cells were treated with or without 100 ng/mL LPS for 6 h. After washing 3 times, AtT-20 cells were added to the lower chamber. Simultaneously, neutrophils, monocytes, or BMDMs ($10^5$ cells) suspended in 100 μL medium were added to the upper chamber. The cells were then allowed to migrate through the insert membrane for 6 h at 37˚C in a 5% $CO_2$ atmosphere. For adherent BMDMs, the inserts were washed with DPBS for 3 times, and non-migrating BMDMs remaining on the upper surface were cleaned with a cotton swab. The migrated BMDMs on the insert were fixed with PFA, stained with DAPI, and counted under a microscope. Non-adherent neutrophils and monocytes were distinguished from AtT-20 cells and counted by FACS after surface marker staining (CD11b for monocytes and Ly6G for neutrophils).

## FACS

The pituitary tissue was collected after treatment and digested into a single-cell suspension with collagenase type II and type IV enzymes. The digested cell suspension was centrifuged, and the pelleted cells were resuspended and filtered through a 75-μm cell strainer. After

centrifugation, the cell pellets were resuspended in RBC lysis buffer (R1010, Solarbio) for 5 min. After washing with DPBS, the collected cell pellets were resuspended in Fixable Viability Dye-eFluor 506 (65-0866-14, Thermo Scientific). After blocking with anti-CD16/32 (101302, BioLegend), the cells were stained with antibodies diluted in FACS buffer (2 mM EDTA and 2% FBS in DPBS) for 20 min at 4°C. For CD206 intracellular staining, a Fixation/Permeabilization Kit (00-5523-00, Invitrogen) was used according to the manufacturer's protocol. Cells were then subjected to flow cytometry analysis using FlowJo software (V10). For blood cell analysis, mice were given 1 µg rNptx2 (7816-NP-050, R&D system) intravenously and i.p. injected 0.5 mg/kg LPS, and whole blood was collected from the orbit. Blood (100 µL) was mixed with 50 µL 8 mM EDTA (in DPBS) and then resuspended in 1.5 mL RBC lysis buffer for 5 min. After washing with DPBS, the cell pellet was stained as described above. The following antibodies were from BioLegend and the dilutions were 1:200: anti-F4/80-PE (Cat: #123109), anti-CD45-APC/Cyanine7 (Cat:#103115), anti-CD11b-FITC (Cat:#101206), anti-Ly6C-Brilliant Violet 605 (Cat:#128035), anti-CD206 (MMR)-PE/Cyanine7 (Cat:#141719), anti-CD11c-APC/Cyanine7 (Cat:#117323), anti-Ly6G-PerCP/Cyanine5.5 (Cat:#127616), anti-CD3-APC (Cat:#103115), anti-CD45R/B220-Pacific Blue (Cat:#103230).

## Immunoblot

Tissue extracts for protein analysis were homogenized using TGrinder (OSE-Y30, TIANGEN) in RIPA buffer (50 mM Tris-HCl (pH 7.5), 450 mM NaCl, 1% NP-40, 0.1% SDS, 0.5% deoxycholic, and cocktail protease inhibitors). Total protein was separated by SDS-PAGE on 10% acrylamide/bisacrylamide gels and transferred to PVDF membranes (1620177, Bio-Rad). Membranes blocked with protein blocking buffer were first incubated with the indicated primary antibodies, followed by horseradish peroxidase-conjugated anti-rabbit IgG or anti-mouse IgG secondary antibody. The following antibodies and dilutions were used: anti-Nptx2 (1:1,000 dilution, 10889-1-AP, Proteintech), anti-β-Actin (1:4,000 dilution, A5441, Sigma-Aldrich), goat anti-mouse IgG: HRP (1:30,000 dilution, 31430, Thermo Fisher Scientific), and goat anti-rabbit IgG: HRP (1:30,000 dilution, 31460, Thermo Fisher Scientific).

## Macrophage depletion and virus injection in the pituitary

Mice were anesthetized with pentobarbital (80 mg/kg, i.p.) before surgery and then placed in a mouse stereotaxic instrument. Injections were performed using a microsyringe pump and a Micro4 controller (World Precision Instruments). For macrophage depletion, liposome-PBS or liposome-Clodronate (ClodronateLiposomes.com) was stereotaxically microinjected into the anterior pituitary (2.5 mm posterior from Bregma, 0.4 mm lateral, 6 mm below pia). The liposomes were delivered to the target site at a rate of 60 nL/min for 500 nL per site. Mice received saline or 0.5 mg/kg LPS 18 h after liposome delivery and were killed 6 h after inflammation was established. For CCL2 expression and Nptx2-KO, AAV was delivered directly into the pituitary. The injection site, rate, and volume were the same as those used for the liposome injection. Subsequent experiments were performed at least 3 weeks after virus injection.

## Sequence data preprocessing, quality control, and analysis

UMI_tools were used for cell barcode and UMI extraction and demultiplexing. STAR (v2.7.9a) was used for sequence alignment. Subread featureCounts (v1.6.3) was used for feature assignment. Genome index was created with GRCm38 mm10 build, downloaded from RefSeq (https://www.ncbi.nlm.nih.gov/refseq/). For quality control, we computed library size, number of genes recovered, mitochondrial RNA percentage, and Ribosomal RNA percentage from the

gene-by-cell expression matrix. The R package scater was used to detect and filter outliers based on the median absolute deviation (MAD) of these metrics.

For scRNA-seq data analysis, Seurat was used to perform normalization, dimensional reduction, integration, and label transfer on the dataset using default parameters. SCENIC was used to infer the regulon activity matrix. Differential expression analysis was performed using the MAST algorithm. Augur was used to prioritize HPC cell types based on their responses to immune challenges. The specificity of gene expression in HPCs was calculated with a previously described method [75]. GO analysis was performed using clusterProfiler. Cell–cell communication was analyzed with CellChat. Data visualization used Seurat, Scanpy, ggplot2, and UpSetR.

## Supporting information

**S1 Fig. Robust similarity in the pituitary response to endotoxin and bacterial-induced inflammatory.** (**A**) Venn diagrams showing the intersection of up-regulated (left panel) and down-regulated (right panel) DEGs in the pituitary under medium-dose (5 mg/kg) LPS (i.p.) treatment and mid-grade CLP treatment for 6 h, and the control groups treated with saline (i. p.) or sham surgery. (**B**) PCA of the pituitary transcriptomes from mice subjected to LPS, CLP for 6 h, and the control groups as indicated in (**A**) ($n$ = 3 replicates). (**C**) GSEA profiles showing significant enrichment of gene sets after CLP treatment in the pituitary. The transcriptome datasets used in (**A**) were used for GSEA analysis. The data underlying this figure can be found in S1 Table. CLP, cecal ligation and puncture; PCA, Principal component analysis; GSEA, Gene Set Enrichment Analysis.
(TIFF)

**S2 Fig. scRNA-seq data quality control (QC) metrics, the binarized SCENIC regulon activity and correlation with bulk RNA data.** (**A–C**) Quality control metrics showing the number of genes recovered per cell and UMI counts recovered per cell (**A**), mitochondrial RNA fraction (**B**), and ribosomal RNA fraction (**C**). Dots marked in red are cells that failed to pass the QC metrics and were removed from subsequent analyses. (**D**) Cell type composition of the sequenced pituitary single cells. (**E**) Violin plots showing expression of canonical marker genes for identified pituitary cell types as indicated in (**Fig 1E**). (**F**) Heatmap showing the 356 binarized SCENIC regulon activity in HPCs with healthy and inflammatory cell states predicted in (**Fig 1I**). Columns are cells and rows are regulons. White: not activated; black: activated. (**G**) Scatterplots showing the correlation between scRNA-seq of 6 HPCs and bulk RNA-seq of the pituitary under LPS treatment. Genes up-regulated or down-regulated by more than 2-fold in bulk RNA-seq are indicated in orange and blue, respectively. Dashed line represents the fitting curve. The data underlying this figure can be found in S2 and S4 Tables.
(TIFF)

**S3 Fig. Imputation of state for the pituitary HPCs in Poly(I:C) and TNF-α treatment groups.** (**A**) Bar plot showing the predicted states of HPCs in Poly(I:C) and TNF-α groups. (**B**) Joint embedding of pituitary single cells using UMAP, with the query dataset [Poly(I:C) and TNF-α] projected onto the reference structure [Saline and LPS treatments]. (**C** and **D**) Visualization of all cell types (**C**) and treatments (**D**) in the query dataset, using the same UMAP embedding as in (**B**). (**E**) Alluvial plot showing the cell state distribution of 6 HPCs under Poly (I:C) and TNF-α treatment groups. HPCs, hormone-producing cells.
(TIFF)

**S4 Fig. Conserved marker genes in 6 HPCs under inflammation.** (**A–F**) UMAP plots showing conserved marker genes for pituitary HPCs. *Car10* is used as a conserved marker for

somatotropes (**A**), *Edil3* for lactotropes (**B**), *Palld* for corticotropes (**C**), *Megf11* for melano-tropes (**D**), *Ttc24* for gonadotropes (**E**), and *Pvalb* for thyrotropes (**F**). HPCs, hormone-producing cells.
(TIFF)

**S5 Fig. REVIGO analysis of 6 HPCs.** (**A–F**) REVIGO analysis showing the interactive graph of 6 HPCs under systemic inflammation. Bubble color indicates the *p*-value. Bubble size indicates the frequency of the GO term in the underlying GOA database. (**G**) Summary shared and unique cell signaling pathways in HPCs from (**A–F**). The representations were created with BioRender.com. REVIGO analysis URL (http://revigo.irb.hr). The data underlying this figure can be found in S9 Table. GOA, Gene Ontology Annotation.
(TIFF)

**S6 Fig. The specific up- and down-regulated GO pathways in corticotrope under systemic inflammation.** (**A and B**) GO analysis on unique up-regulated (**A**) and down-regulated (**B**) DEGs of corticotropes. (**C–F**) UMAP plots showing up-regulated DEGs in corticotropes. (**G–J**) UMAP plots showing down-regulated DEGs in corticotropes. The data underlying this figure can be found in S10 Table.
(TIFF)

**S7 Fig. Systemic inflammation changes secretory functions in the pituitary.** (**A**) Violin plots showing the expression of canonical HPCs marker genes under healthy or inflammatory states. Violin line color: cell states. (**B**) Volcano plot showing the pituitary proteome for differentially abundant proteins in control and LPS (50 mg/kg LPS for 6 h) groups. (**C**) GO analysis on up-regulated proteins from (**B**). (**D**) Violin plots showing the expression of myeloid migration-related chemokine genes under healthy or inflammatory states. Violin line color: cell states. The data underlying this figure can be found in S12 Table. HPCs, hormone-producing cells.
(TIFF)

**S8 Fig. The pituitary secretes Nptx2 under systemic inflammation.** (**A–C**) UMAP plots showing DEGs identified from the scRNA-seq dataset. (**D**) Representative images showing ISH of *Crhr1* RNA (green) and IF of POMC (red) in the pituitary from mice treated with saline or LPS for 6 h. Scale bar, 50 μm. (**E and F**) UMAP plots showing *Nptx2* (**E**) and *Cartpt* (**F**) from the scRNA-seq dataset. (**G**) qPCR analysis of *Nptx2* in the pituitary from mice treated with LPS (0.5 mg/kg) for different durations (*n* = 3–4 mice). (**H**) Histogram showing the statistic information of the percentage of *Nptx2*+ cells in POMC+ cells. (**I**) Immunoblot analysis of Nptx2 in the pituitary from LPS (0.5 mg/kg LPS exposure for 6 h)-challenged mice 3 weeks following the infusion of AAV-SaCas9-U6-sgRNA-*Nptx2* or the control vectors into the pituitary (*n* = 3 or 4 mice per group). (**J**) qPCR analysis of *Nptx2* in AtT-20 cells after treatment with control or inflamed serum for 6 h (*n* = 6 replicates). (**K**) qPCR analysis of *Nptx2* in AtT-20 cells after treatment with PBS, 200 ng/mL LPS, or 100 ng/mL cytokines for 6 h (*n* = 3 replicates). All data represent mean with SD. Statistical significance was determined by two-tailed Student's *t* test, *$p < 0.05$, **$p < 0.01$, ***$p < 0.001$, ****$p < 0.0001$. The data underlying this figure can be found in S1 Data. The original blot for this blot can be found in S1 Raw Images. ISH, in situ hybridization; IF, immunofluorescence.
(TIF)

**S9 Fig. Immune cells are recruited to the pituitary through HPCs during systemic inflammation.** (**A**) qPCR analysis of several major chemokine-encoding genes in AtT-20 cells after treatment with PBS, 200 ng/mL LPS, or 100 ng/mL cytokines for 6 h (*n* = 3 replicates). (**B**)

qPCR analysis of the chemokine-encoding genes in AtT-20 cells after treatment with control or inflamed serum for 6 h ($n$ = 3 replicates). (**C**) FACS gating strategies for neutrophil and microglia/macrophage. (**D**) FACS analysis for macrophages (red circle) among Ly6G⁻ cells in the pituitary from mice treated with LPS at different times. (**E**) Representative images showing IF of CD31 (purple, the marker of endothelial cell) and Iba1 (green) in the pituitary from mice treated with saline or LPS. Scale bar, 100 μm. (**F**) FACS analysis for neutrophils (red rectangle) among CD45⁺ cells in the pituitary from mice treated with LPS at different times. (**G**) Representative images showing IF of CD31 (purple) and Ly6G (green, the marker of neutrophil) in the pituitary from mice treated with saline or LPS. Scale bar, 100 μm. (**H**) qPCR analysis of the representative macrophage marker gene *Cd11b* in the pituitary from mice treated with LPS. (**I** and **J**) FACS analysis of CD206⁺ M2-like macrophages (**I**) and the ratio of CD206⁺/CD11c⁺ macrophages (**J**) in the pituitary from mice treated with LPS ($n$ = 3 mice). (**K** and **L**) qPCR analysis of the M2-like macrophage marker genes *Mrc1* (**K**) and *Cd301* (**L**) in the pituitary from mice treated with LPS. The injection dose of LPS was 0.5 mg/kg in (**C–L**). All data represent mean with SD. Statistical significance was determined by two-tailed Student's *t* test, *$p < 0.05$, **$p < 0.01$, ***$p < 0.001$, ****$p < 0.0001$. The data underlying this figure can be found in S1 Data. IF, immunofluorescence.
(TIF)

**S1 Raw Images. Original blots used in Figs 3F and S8I.**
(PDF)

**S1 Data. Excel spreadsheet containing the numerical data for Figs 1D, 3D, 3G–3L, 4E–4G, 5A, 5B, 5D–5H, S8G, S8H, S8J, S8K, S9A, S9B and S9H–S9L.**
(XLSX)

**S1 Table. TPM for bulk RNA-seq as depicted in Figs 1B, 1C, and S1.**
(CSV)

**S2 Table. Metadata table of single cells showing quality control metrics, experimental conditions and cell type, cell state annotations as depicted in S2A–S2C Fig.**
(CSV)

**S3 Table. Top 2,000 highly variable genes identified from single pituitary cell transcriptomes for Fig 1G.**
(CSV)

**S4 Table. 356 SCENIC identified regulons for dimension reduction as depicted in S2F Fig.**
(CSV)

**S5 Table. Conserved markers of pituitary HPCs across inflammation and healthy conditions as depicted in Fig 2B.**
(CSV)

**S6 Table. Differentially expressed markers of pituitary HPCs between inflammation and healthy conditions as depicted in Fig 2B.**
(CSV)

**S7 Table. Venn relationship of conserved markers across pituitary HPCs as depicted in Fig 2C.**
(CSV)

**S8 Table. Venn relationship of differentially expressed markers across pituitary HPCs as depicted in Fig 2D.**
(CSV)

**S9 Table. List of GO pathways used for REVIGO analysis as depicted in S5 Fig.**
(XLSX)

**S10 Table. List of corticotrope-specific DEGs used for GO analysis as depicted in S6 Fig.**
(XLSX)

**S11 Table. List of DEGs used for transcriptome level GO analysis as depicted in Fig 3A and 3B.**
(CSV)

**S12 Table. List of genes used for proteome level GO analysis as depicted in S7B and S7C Fig.**
(CSV)

**S13 Table. Primer sequences used in this study.**
(XLSX)

## Acknowledgments

The authors thank all members of ML's laboratory for their assistance in this study. We thank Fuchou Tan and Yueli Cui (Peking University) for their assistance in scRNA-seq experiments. We thank Chenxi Jia and Di Yao (National Center for Protein Sciences-Beijing) for their assistance in the pituitary proteomics experiments. We thank Tao Cai (National Institute of Biological Sciences, Beijing) for his assistance in bioinformatics analysis. We thank Rui Lin (National Institute of Biological Sciences, Beijing), Xiangyu Li (currently School of Software Engineering, Beijing Jiaotong University), and Hongjun Li (Michael Q. Zhang Lab, TNLIST Bioinformatics Division, Tsinghua University) for discussions.

## Author Contributions

**Conceptualization:** Ting Yan, Ruiyu Wang, Minmin Luo.

**Data curation:** Ting Yan, Ruiyu Wang, Jingfei Yao.

**Formal analysis:** Ting Yan, Ruiyu Wang.

**Funding acquisition:** Minmin Luo.

**Investigation:** Ting Yan, Ruiyu Wang, Jingfei Yao.

**Methodology:** Ting Yan, Ruiyu Wang, Jingfei Yao.

**Project administration:** Ting Yan, Minmin Luo.

**Resources:** Minmin Luo.

**Software:** Ruiyu Wang.

**Supervision:** Minmin Luo.

**Validation:** Ting Yan, Ruiyu Wang.

**Visualization:** Ting Yan, Ruiyu Wang, Jingfei Yao.

**Writing – original draft:** Ting Yan, Ruiyu Wang, Minmin Luo.

**Writing – review & editing:** Ting Yan, Ruiyu Wang, Minmin Luo.

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
