## [Editor Report · Decision Letter 0]

24 Mar 2023

Dear Dr Luo, 

Thank you for submitting your manuscript entitled "Single-cell Transcriptomic Analysis Reveals Rich Pituitary-immune Interactions under Systemic Inflammation" for consideration as a Methods and Resources article by PLOS Biology.

Your manuscript has now been evaluated by the PLOS Biology editorial staff as well as by an academic editor with relevant expertise and I am writing to let you know that we would like to send your submission out for external peer review.

Once your full submission is complete, your paper will undergo a series of checks in preparation for peer review. After your manuscript has passed the checks it will be sent out for review. To provide the metadata for your submission, please Login to Editorial Manager (https://www.editorialmanager.com/pbiology) within two working days, i.e. by Mar 28 2023 11:59PM.

Kind regards,

Lucas

Lucas Smith, Ph.D.

Associate Editor

PLOS Biology

lsmith@plos.org

---

## [Decision Letter · Decision Letter 1]

19 May 2023

Dear Dr Luo,

Thank you for your patience while your manuscript "Single-cell transcriptomic analysis reveals rich pituitary-immune interactions under systemic inflammation" was peer-reviewed at PLOS Biology, and apologies for the protracted review process. Your study has now been evaluated by the PLOS Biology editors, an Academic Editor with relevant expertise, and by several independent reviewers. 

In light of the reviews, which you will find at the end of this email, we would like to invite you to revise the work to thoroughly address the reviewers' reports.

As you will see below, the reviewers have commented that the study is interesting, but they have raised a number of comments which we think should be addressed before we can consider your manuscript for publication. Multiple reviewers have commented that the manuscript is organized in a way that was difficult to follow - and we think that the manuscript should be carefully revised to improve this aspect (although we will leave it optional to remove parts of the paper, as Reviewer 2 suggests, as we think the study is broadly within the scope of our 'Resource' article type). We think it would be important to add the control experiment suggested by Reviewer 1 and to perform the different analyses suggested by Reviewer 2 which will strengthen the study.

As a note, Reviewer 3 suggested an experiment to modulate neutrophils, either with depletion or blocking antibodies. While we think this is a very interesting line of study, we would not strictly require this data, as we feel it may be beyond the scope of this paper.

Given the extent of revision needed, we cannot make a decision about publication until we have seen the revised manuscript and your response to the reviewers' comments. Your revised manuscript is likely to be sent for further evaluation by all or a subset of the reviewers.

**IMPORTANT - SUBMITTING YOUR REVISION**

*Re-submission Checklist*

*Published Peer Review*

*PLOS Data Policy*

*Blot and Gel Data Policy*

Sincerely,

Lucas

Lucas Smith, Ph.D.

Associate Editor

PLOS Biology

lsmith@plos.org

REVIEWS:

Reviewer #1, Qingyun Li (Note, reviewer 1 has signed this review) : In this manuscript, Yan et al. leverage single-cell RNA sequencing (scRNAseq) to investigate how pituitary hormone-producing cells (HPCs) respond to proinflammatory stimuli. HPCs along the HPA axis play a key role in modulating physiological functions of the body in both homeostasis and disease. Previous scRNAseq studies have characterized the transcriptional landscape of the pituitary under healthy conditions. This work distinguishes itself by demonstrating changes in the six major types of HPCs in response to LPS, Poly I:C or CLP, with corticotropes displaying the strongest reaction. This discovery in itself is quite significant. What's really impressive about this work is that the authors go on and provide mechanistic insights into how pituitary cells interact bidirectionally with immune cells to modulate protein secretion (either immune- or hormone- related). These functional studies lead to the discovery of Nptx2 as a novel pituitary-specific factor regulating neutrophil-leukocyte balance, as well as the function of macrophages in triggering ACTH secretion upon proinflammatory stimulation. Overall, the presented data is of high quality, the delivery is clear and concise, and the take-home message is exciting. Please consider the following comments for revision. 

1. The infiltration of neutrophils was demonstrated using flow cytometry which lacks spatial information and does not rule out the possibility of blood contamination or the accumulation of these cells at the brain borders. If these cells infiltrated into the parenchyma, were they only found in the pituitary? It is recommended to provide histology data to complement this finding. 

2. The section on Crhr1 is a bit confusing as some of the provided data suggests that LPS causes an increase in hormone secretion, as evidenced by the enrichment of the "positive regulation of secretion" term in DEGs, with Crhr1 being one of the representative genes (line 180). However, the downregulation of Crhr1(Fig S5H) in the LPS treated group seems to contradict this finding. The authors should clarify the interpretation of this data. Additionally, it would be helpful to provide clearer information about the directionality of regulation on Line 185-186. 

3. On Line 200, the authors mentioned proteomic analysis but did not provide a description of how it was performed. It would be helpful to include more information about the methods used for this analysis. Additionally, it is not clear whether the cytokines/chemokines that show upregulation in the transcriptomic data are also detected in the proteomic dataset. Clarification on this point would be valuable. 

4. The terminology for M1/M2 macrophage is considered to be obsolete, and it may be more appropriate to use marker genes such as Mrc1+ to describe the population. This would be more reflective of the current understanding of macrophage polarization. 

Reviewer #2: I have read with interest the manuscript entitled "single-cell transcriptomic analysis reveals rich pituitary-immune Interactions under systemic inflammation". The manuscript is generally well-written in Standard English. The high number and the complexity of techniques used in this study are quite impressive. 

The major advantage and novelty of this study is an analysis of individual transcriptomic responses of pituitary HPCs (hormone-producing cells) to diverse experimental inflammatory conditions such as those induced by systemic administration of TNF, LPS, Poly I:C, and CLP. Results of this study could therefore be potentially used in the identification of biomarkers of sepsis-induced pituitary dysfunction. In fact, an example of a potential biomarker, namely NPTX2 has been found and proposed based on the results of this study. 

However, the study design is overcomplicated, some experiments confusing or unnecessary, and the manuscript seems to be in some parts defocused. Because of that, the manuscript reads as 2-3 separate papers, with different degrees of finalization. 

Here are my remarks:

1. The study starts with a bulk RNAseq analysis of the pituitary transcriptome changes during LPS- and CLP-induced-mediated inflammation. 

a. Analysis of those experiments provided rather expected results, which are not organ-specific such as apoptosis, and generally pointing to tissue inflammation (increased cytokines and their signaling pathways). Unfortunately, except for the "myeloid leukocyte migration pathway - chemokines", other pathways identified in these experiments were not studied further in a cell-specific manner later.

b. In addition, no clear benefit to the main story of the manuscript came from the comparison of LPS with CLP experiments (meaning this part could also be removed). Unless providing additional data would be considered. For example, it would be of interest to find similarities and differences between the top DEG (differentially expressed genes) in pituitaries of LPS- and CLP-treated mice. In the end, LPS-driven systemic inflammation is TLR-4 dependent whereas CLP-induced peritonitis, a more physiological mouse model of sepsis, is driven by whole bacteria activating multiple TLRs and other PRRs. Afterward, those results could be compared to scRNA seq results from different subsets of pituitary HPCs.

c. There are also only four pathways presented in both conditions. Are they no other significantly enriched gene sets (pathways) in both groups (CLP & LPS)? It has been shown that elevated GCs concentrations can negatively influence the pituitary function through e.g. negative feedback induction, suppression of PC1/3, which enzyme process POMC into ACTH, and via suppressing ACTH release through upregulation of Annexin A1. Therefore, it would be of interest to see whether there are these or other GC-regulated genes (DEG) present in those datasets (e.g. Gliz, etc). 

d. What was the rationale for using the 6h time point? Usually, for gene expression, 2-3 hours are sufficient and mark the top of transcriptomic responses. Initial bell-like response (max expression) after LPS injection is rather expected, however, it usually occurs at 2-3-hour time points. Therefore, I wonder why this time was not included in the analysis.

e. Besides p-values, GSEA pathways should also contain information about NES (nominated enrichment scores) and FDR (false detection rates). Please provide also more details about your CLP procedure in the Materials and Methods section.

2. In order to analyze an individual response of the pituitary HPCs to inflammation, a sc RNA seq was performed on pituitary cells isolated from mice undergoing either LPS- (conc. 0.5-50 mg/kg & time 3-504 hrs. "3wks"), poly I:C- or TNF-alpha- induced systemic inflammation. Results of this study demonstrated a clear separation of different HPCs within pituitary gland microenvironment and indicated "healthy" and "inflamed" status of different pituitary HPCs subsets. Moreover, it indicated that corticotropes and somatotrophs showed the highest transcriptomic responses during systemic inflammation. 

a. Based on which DEG sets were those two cellular states defined? proinflammatory cytokines, chemokines or other factors? Were the DEGs sets used to characterize "inflammatory state" the same among different pituitary HPCs subsets? Meaning the same cytokines/chemokines etc. This needs to be better clarified and depicted in the figure. 

b. Considering differences in the inflammatory responses induced by LPS, poly I:C- and TNF-alpha, it would be of great interest to analyze and compare the number of DEGs and their interactions (GSEA, etc.) individually and between each other in each pituitary HPCs subsets. 

c. The analysis of how different concentrations of P IC influence the number of healthy vs. inflamed cells does not bring much information (Fig.S3). 

3. Figure 3 of the manuscripts shows a number of conserved and differentially expressed "marker" genes from each pituitary HPC set. The expression of the latter genes was then compared between different organs by qPCR. 

a. I am not sure what identification of the conservative genes in each pituitary HPC subset brings to the main story. Moreover, their conservative status is not clear to me. For example, the Edil3 gene has been identified as a "conservative" gene for lactotrophs. But Del-1 encoded by this gene is the negative regulator of leukocyte adhesion to endothelial cells, through blocking interactions of beta 2 integrin with ICAMs. It is expressed in restricted organs and cell types including the brain, eye, or adrenals, where its expression is strongly suppressed during LPS-induced inflammation. In other words, it is hard to imagine that in the lactotrophs, its expression would not be altered by inflammation. 

b. Next part of the manuscript (2nd story), focuses on the identification of NPTX2 as an inflammatory marker gene of corticotropes. Previously, decreased levels of NPTX2 have been identified as a prognostic biomarker of Alzheimer's disease and schizophrenia (or in general as accelerated cognitive decline). Meanwhile, overexpression of Nptx2 was sufficient to restrain complement activity (C1q) and ameliorate microglia-mediated synapse loss. Those results indeed demonstrate that NPTX2 plays an important role in the "health" status of neuronal cells. 

Interestingly, qPCR data from Fig.3D showing decreased expression of this gene in the cortex and midbrain seems to be in favor of the above data. 

c. However, the question remains whether actually, pituitary gland is the only "sole" contributor of NPTX2 in the plasma during inflammation. Often mRNA changes at a given time may not necessarily reflect the actual serum protein levels. For example, expression of the Nptx2 gene might have been induced at the earliest time point in certain organs. Were all data (including qPCR) shown in Fig. 3 driven from 6 hours time point after LPS injection? If yes, it could have been stated as 1 sentence at the end of Fig.3's legend. 

Moreover, looking at Fig. 3E, it seems that although the Nptx2 gene expression was found in Pomc expressing cells, and it increased a bit after LPS injection (0.5 mg/kg), it did not markedly change when higher concentration (inflammation) was tested (40 mg/kg). It doesn't correspond to WB results showing a strong upregulation of NPTX2 in the pituitary after 50 mg/kg of LPS. Of interest, there is no information about 40 mg/kg of LPS treatment in the legends of Figure Was it 40 or 50 mg/kg? 

d. As presented in Fig. S7G serum NPTX2 decreased partly after silencing of the Nptx2 gene in the pituitary suggesting that indeed pituitary gland might contribute to its systemic levels, however, it was not fully alleviated. Thus, NPTX2 measurement after LPS injections into hypophysectomized mice could provide a valuable/more direct hint, as to whether this protein indeed comes only mostly from the pituitary gland. Was serum NPTX2 concentration also elevated after TNF or Poly I: C injection? Of interest, do AtT20 cells also express NPTX2 and can elevate it after LPS or TNF treatment? That would be an easier experiment to do instead of hypophysectomy.

e. I am quite confused why NPTX2 levels were connected to PBMC composition and not for example with pituitary gland damage. Or the nerve synapses in the pituitary. Alternatively, I would definitely expect some connection with GC effects. It is also weird that rNPTX2 alone leads to a decrease in T cells % of PBMC whereas LPS did not (Fig.3I). Moreover, it should be clarified whether injection of rNPTX2 in fact modulates GC-exerted effects on PBMC composition. Did rNPTX2 injection alter GC levels in LPS-treated mice? 

4. To my opinion the CellChat results and overexpression of Ccl2 does not bring 

anything to the manuscript except of the its title. Chemokines are known to be secreted form different inflamed tissues and they would attract specific immune cells. 

5. The last part of the manuscript (macrophages promote ACTH release under 

inflammation) is the third independent part of the manuscript (meaning it has been partially known before, and the experiments presented in the Fig. 5 just nicely support it. Those results can be used for an independent manuscript as they have little connection with previous 2 individual parts of the manuscript. 

Minor

Line 62: sentence is not clear. Especially considering "mechanisms of pituitary HPC"

Line 63-4: Unclear Aim. There is a direct immune-HPC interactions mentioned as an aim of study. Which experiments performed in this manuscript had actually addressed it? 

Reviewer #3: In the paper by Yan et al, the authors use single cell transcriptomic to understand how hormones producing cells respond to systemic inflammation. Overall, the paper is novel and interesting. Some improvement would however enhance the readability and impact of the manuscript.

Major Points:

- Overall the organization of the manuscript make is sometimes hard to follow. Particularly, entire subpart of the manuscript are referring entirely to supplementary figures (Supp Fig1, Supp Fig6). Given the importance of the finding discussed in those figures, they should be brought up into the main figures. The authors demonstrate that chemokine are strongly unregulated in HPCs, then move on to NPTX2 to come back to chemokines with the last 2 figures. In this reviewer's opinion, the chemokine story should be one part before moving on to the NPTX2 study. The NPTX2 study is in this reviewer's opinion the most innovative piece of the study in this manuscript, yet some of the important data is placed in a supplementary figure. Some figure and paper re-organization would greatly streamline the message the author are putting forward.

- In the last 2 figures, the authors aim that "inflamed" HPC produce chemokines to recruit immune cells that then would act on the HPC to produce more hormones. The data as currently presented is circumstantial and the experimental design has some flaws that hinder proper interpretation of the results. In Figure 4, the authors show an up regulation of chemokines by HPC upon inflammation that is particularly linked to neutrophils attraction. The authors go on to try to demonstrate immune cell recruitment into the pituitary gland upon LPS challenge. The gating strategy used by the author to discriminate resident macrophages from recruited is based on CD11b expression levels. This reviewer's question the validity of this approach. Other markers should be use to validate that this strategy discriminate between microglia and infiltrating macrophages. The gating strategy and markers to identify neutrophils should also be provided. The authors refers to M1 and M2 markers. The macrophage field has now moving away from this nomenclature that fails to illustrate the actual dynamic of macrophage phenotype. The author should refrain from using these terms. The authors should validate the immune infiltration into the pituitary using histology to really demonstrate the intraparenchymal infiltration upon LPS challenge. In the last figure, the author use local clodronate liposomes to assess how "macrophages" recruitment impact ACTH production upon LPS challenge. As denoted in Figure 5C, the approach depletes infiltrating macrophages (along with other peripheral immune cells) but also microglia. This caveat should be disclosed and discussed in the manuscript as it can be a strong confounding factor in the interpretation of their results. Does clodronate liposomes affects ACTH production at baseline ? If so this would suggest that microglia are essential for ACTH production by HPC. Figure 4 emphasize how neutrophils seem to be a primary target, but the author move on to macrophages in Figure 5. While challenging, depletion of neutrophils has been demonstrated by multiple group. The authors could also modulate neutrophils recruitment by blocking adhesion molecule on the blood vasculature, or block some of the identified chemokines using commercially available blocking antibodies. These experiments, coupled with the CCL2 experiments already presented would greatly strengthen this part of the manuscript. Have the authors look if macrophages/neutrophils recruitment are also the driver for NPTX2 to serve as a negative feedback loop to limit uncontrolled inflammation ? Attempt to link the two majors parties of this manuscript would be fantastic, although not required to find this manuscript suitable for publication. 

Minor Points:

- The IHC data presented in figure 3 and Supp figure 5 are hard to read, even on a computer. The addition of zoomed in inset would greatly help the reading of these figures. 

- In figure S4G, the author should also provide the statistical representation of the pathways to reinforce that these were not randomly chosen and rather where mathematically shown to be important. 

- Figure 1H is extremely hard to read and interpret, and seem redundant with Figure 2B. The authors may consider another representation of the results that is easier to read or remove it completely.

- On page 4, the authors refer to TNFa injection as a "cytokine storm". I would encourage the authors to avoid using this term as Cytokine Storm is defined by the simultaneous massive upregulation of numerous cytokines.

---

## [Decision Letter · Decision Letter 2]

9 Oct 2023

Dear Dr Luo,

Thank you for your patience while we considered your revised manuscript "Single-cell transcriptomic analysis reveals rich pituitary-immune interactions under systemic inflammation" for publication as a Methods and Resources article at PLOS Biology. This revised version of your manuscript has been evaluated by the PLOS Biology editors, the Academic Editor and the original reviewers.

As you will see in their comments below, the reviewers are largely satisfied by the revision, and two reviewers have suggested we accept the paper. However Reviewer 3 has two lingering concerns that we think should be addressed before publication. We are likely to accept this manuscript for publication, provided you satisfactorily address the remaining points raised by reviewer 3. To that end, we agree with Reviewer 3 that the data from the Response to Reviewers should be added to the manuscript. Regarding Reviewer 3's second point, we think this can be addressed by either i) removing the conclusion that they are infiltrating macrophages ii) providing more stainings to support this claim. 

**IMPORTANT: As you address Reviewer 3's last requests please also make sure to address the following data and other policy-related requests:

1) ABSTRACT: Please note that per journal policy, the model system/species studied should be clearly stated in the abstract of your manuscript. 

2) ETHICS STATEMENT: Please update the ethics statement in your methods section to include an approval number for your animal use protocol, approved by the IACUC from National Institute of Biological Sciences, Beijing. 

3) DATA: Thank you for providing the data underlying your figures as a supplementary excel file and as a deposition to NGDC. 

>> Please can you add a sentence to each relevant figure legend (including supplemental), directing readers to these datasets? 

>> I also had trouble accessing the data provided on NGDC via the accession number provided (PRJCA015861). Can you please provide me a reviewer token so that I can double check that this data meets our requirements?

4) WESTERN BLOTS: Thank you for providing the raw western blots as a supplementary file. Some of these appear to be cropped. Can you please update this file to provide the completely uncropped versions of your western blots? This is needed to be compliant with our blot and gel reporting requirements: https://journals.plos.org/plosbiology/s/figures#loc-blot-and-gel-reporting-requirements

5) CODE: Per journal policy, if any code was generated to support the conclusions of your manuscript, we would require that you make it available without restrictions upon publication. Please ensure that any code is sufficiently well documented and reusable, and that your Data Statement in the Editorial Manager submission system accurately describes where your code can be found.

We expect to receive your revised manuscript within two weeks. 

*Published Peer Review History*

*Press*

Sincerely,

Luke

Lucas Smith, Ph.D.

Senior Editor,

lsmith@plos.org,

PLOS Biology

Reviewer remarks:

Reviewer #1: The authors have fully addressed all of my concerns. 

Reviewer #2: I have no further comments

Reviewer #3: First I would like to thank the authors for addressing the majority of my concerns.

Some concerns however remains.

- Some new figures that are provided to the reviewers in the point-by-point section do not appear to have been included in the revised manuscript (R17, R20, R21...)

- Regarding figure R21, the description by the authors is suggesting that their interpretation of Iba1 staining in close proximity to blood vasculature demonstrates macrophages infiltration? Mutliple reports, includind 2 photon microscope (Davalos et al, 2012) demonstrate that resident microglia will migrate toward blood vessels during inflammation. The use of multiple markers (Iba1, Tmem119, ...) would help the authors differentiate between microglia and infiltrating macrophages.

---

## [Editor Report · Decision Letter 3]

26 Oct 2023

Dear Minmin,

Thank you for the submission of your revised Methods and Resources "Single-cell transcriptomic analysis reveals rich pituitary-immune interactions under systemic inflammation" for publication in PLOS Biology, and thank you for addressing the last reviewer and editorial requests in this revision. On behalf of my colleagues and the Academic Editor, Richard Daneman, I am pleased to say that we can in principle accept your manuscript for publication, provided you address any remaining formatting and reporting issues. These will be detailed in an email you should receive within 2-3 business days from our colleagues in the journal operations team; no action is required from you until then. Please note that we will not be able to formally accept your manuscript and schedule it for publication until you have completed any requested changes.

**IMPORTANT: As you address the formatting and reporting requests to come, please also address the following editorial request: 

1) Thank you for updating your S1_raw images file to include the uncropped western blots related to your figures. As indicated in our discussion over email, in cases where you have added "Xs" to this file, to indicate irrelevant bands, please make sure that the X does not obscure the western blot image. The Xs should be moved up a bit, to the top of each lane. 

PRESS

Sincerely, 

Luke

Lucas Smith, Ph.D.

Senior Editor

PLOS Biology

lsmith@plos.org